# BEYOND REDUNDANCY: DIVERSE AND SPECIALIZED MULTI-EXPERT SPARSE AUTOENCODER

## ABSTRACT

Sparse autoencoders (SAEs) have emerged as a powerful tool for interpreting large language models (LLMs) by decomposing token activations into combinations of human-understandable features. While SAEs provide crucial insights into LLM explanations, their practical adoption faces a fundamental challenge: better interpretability demands that SAEs' hidden layers have high dimensionality to satisfy sparsity constraints, resulting in prohibitive training and inference costs. Recent Mixture of Experts (MoE) approaches attempt to address this by partitioning SAEs into narrower expert networks with gated activation, thereby reducing computation. In a well-designed MoE, each expert should focus on learning a distinct set of features. However, we identify a *critical limitation* in MoE-SAE: Experts often fail to specialize, which means they frequently learn overlapping or identical features. To deal with it, we propose two key innovations: (1) Multiple Expert Activation that simultaneously engages semantically weighted expert subsets to encourage specialization, and (2) Feature Scaling that enhances diversity through adaptive high-frequency scaling. Experiments demonstrate 24% lower reconstruction error and reduced feature redundancy compared to existing MoE-SAE methods. This work bridges the interpretability-efficiency gap in LLM analysis, allowing transparent model inspection without compromising computational feasibility. Our code is publicly available in an anonymous repository.

## 1 INTRODUCTION

The rapid advancement of Large Language Models (LLMs) has intensified the need for interpretability to address concerns regarding their safety, reliability, and fairness (Bereska & Gavves, 2024; Zhao et al., 2024; Rai et al., 2025). A primary approach for understanding these models involves analyzing the function of individual neurons. However, this effort is fundamentally hindered by polysemanticity, a phenomenon where a single neuron is activated by multiple, seemingly unrelated concepts, rendering its role ambiguous (Olah et al., 2020). This phenomenon is explained by the superposition hypothesis (Elhage et al., 2022), which posits that polysemanticity arises from the compression of a vast number of real-world features into a finite, high-dimensional activation space. This compression compels individual neurons to represent multiple concepts simultaneously. Consequently, the function of any single neuron becomes entangled and intractable to interpret directly, which fundamentally obstructs our ability to understand the model's internal mechanisms.

Sparse Autoencoders (SAEs) have been proposed to address the challenge of polysemanticity by decomposing model activations into a sparse, monosemantic feature dictionary (Cunningham et al., 2023; Braun et al., 2024; Bussmann et al., 2024). Traditional SAEs employ an overcomplete hidden layer and an $L_1$ penalty to learn this dictionary (Lee et al., 2007; Le, 2013), approximating the decomposition of polysemantic neurons into more interpretable, orthogonal feature directions. This line of research has seen rapid progress, with recent work (Bricken et al., 2023; Rajamanoharan et al., 2024a) successfully scaling SAEs to millions of features and applying them to state-of-the-art models such as Claude 3 Sonnet (Anthropic, 2024) and GPT-4 (Achiam et al., 2023). Despite these advances, the practical adoption of SAEs is hindered by a significant scalability bottleneck (Mudide et al., 2025; Shu et al., 2025). Achieving sufficient sparsity requires an extremely wide hidden layer, resulting in computational costs that grow linearly with the model's hidden dimension. Furthermore, because a trained SAE is layer-specific (Shu et al., 2025), a separate one must be trained for each layer of interest, adding significant computational overhead.

The integration of Mixture of Experts (MoE) architectures into SAEs has emerged as a recent strategy to alleviate this computational cost (Shazeer et al., 2017; Lepikhin et al., 2020; Fedus et al., 2022). These models, exemplified by the Switch SAE (Mudide et al., 2025), partition a large autoencoder into smaller expert subnetworks and dynamically route activations to reduce computational costs while maintaining performance. However, our empirical analysis reveals a fundamental limitation: experts often fail to specialize, collapsing into redundant ensembles where the same features are duplicated across multiple experts. This lack of specialization leads to a recurrence of polysemanticity within each expert; for instance, a single feature might be strongly activated by disparate concepts such as "travel" and "boxer." This expert-level redundancy not only wastes model capacity but also undermines interpretability, as the mapping between learned features and human-understandable concepts becomes fragmented and unclear.

In this work, we introduce Scale Sparse Autoencoder, a dual-mechanism framework designed to enforce both expert specialization and neuron activation diversity. The first mechanism, Multiple Expert Activation, addresses specialization at the specialist level. Instead of activating a single expert, our method dynamically selects and activates a subset of experts for each input. This approach encourages structured specialization by routing the distinct semantic components of a polysemantic neuron to different features within separate experts, which leads each expert to develop a characteristic sensitivity to a particular conceptual domain. The second mechanism, Feature Scaling, operates at the feature level. Inspired by signal decomposition, this technique adaptively amplifies the high-frequency components of the encoder's features in a learnable manner. Our empirical findings show that the model always learns to enhance these high-frequency components. This results in more monosemantic features, an improvement evidenced by a lower similarity to the features and more diverse neuron activation patterns. Extensive experiments validate the superiority of Scale SAE over existing algorithms across three key axes: reconstruction fidelity, functional faithfulness, and feature redundancy. Ablation studies further confirm that these benefits are directly attributable to our two core, mutually reinforcing mechanisms, as ablating either one results in a significant degradation across the aforementioned performance metrics. The primary contributions of this work are as follows.

1. We introduce Scale SAE, a novel framework that integrates two synergistic mechanisms: Multiple Expert Activation and Feature Scaling (Sections 2.2.1 and 2.2.2).

2. We carry out end-to-end experiments showing that Scale SAE substantially outperforms existing methods, achieving a lower reconstruction error, higher Loss Recovered, enhanced interpretability, and reduced feature similarity (Section 3.1).

3. Our ablation studies validate the synergistic contribution of both mechanisms to the model's overall performance (Section 3.2).

4. We provide a detailed mechanistic analysis that explains these performance gains, establishing that Multiple Expert Activation directly improves expert specialization. At the same time, Feature Scaling reduces feature redundancy by activating a more diverse set of features (Section 3.3).

5. We provide a detailed discussion on the limitations of Scale SAE and potential future directions. (Section 4 and Appendix D)

## 2 METHODOLOGY

### 2.1 PRELIMINARY

**Dense SAE.** A traditional sparse autoencoder, such as the TopK variant (Gao et al., 2024), processes input through a single encoder-decoder pair. The training objective is to minimize the reconstruction mean square error (MSE) $\|\mathbf{x} - \hat{\mathbf{x}}\|^2$, where the reconstructed vector is $\hat{\mathbf{x}} = \mathbf{W}^{\mathrm{dec}}\mathbf{z} + \mathbf{b}_{\mathrm{pre}}$ and the sparse code is $\mathbf{z} = \mathrm{TopK}(\mathbf{W}^{\mathrm{enc}}(\mathbf{x} - \mathbf{b}_{\mathrm{pre}}))$.

**MoE SAE.** The MoE SAE architecture improves this design for computational efficiency by employing $N$ distinct "expert" autoencoders. For each input $\mathbf{x} \in \mathbb{R}^{d_{\mathrm{model}}}$, a router selects a single expert $i^*$ based on a learned gating distribution: $i^* = \mathrm{argmax}_i(p_i(\mathbf{x}))$, where $p(\mathbf{x}) = \mathrm{softmax}(\mathbf{W}_g\mathbf{x})$. Each expert can generate an individual reconstruction $E_i(\mathbf{x})$:

$$E_i(\mathbf{x}) = \mathbf{W}_i^{\mathrm{dec}}\mathrm{TopK}(\mathbf{W}_i^{\mathrm{enc}}\mathbf{x}) \tag{1}$$

Figure 1: Scale Sparse Autoencoder Architecture. An illustration of the three core mechanisms in the Scale SAE architecture. **(a) Multiple Expert Activation.** A router selects a subset of experts (e.g., 2 out of 3 shown) to process each input. **(b) Global Top-K Activation.** The activations from the selected experts are aggregated, and a global Top-K operation (K=3 shown) is applied to enforce sparsity. **(c) Feature Scaling.** The encoder weights of each expert are decomposed and scaled to dynamically amplify high-frequency components.

The expert chosen then produces the final reconstruction $\hat{\mathbf{x}} = p_{i^*(\mathbf{x})} E_{i^*(\mathbf{x})}(\mathbf{x} - \mathbf{b}_{\text{pre}}) + \mathbf{b}_{\text{pre}}$.

The total loss function $\mathcal{L}$ combines the reconstruction error $\mathcal{L}_{\text{recon}}$ with an auxiliary load-balancing loss $\mathcal{L}_{\text{aux}}$, weighted by a hyperparameter $\alpha$:

$$\mathcal{L} = \underbrace{\|\mathbf{x} - \hat{\mathbf{x}}\|_2^2}_{\mathcal{L}_{\text{recon}}} + \alpha \cdot \underbrace{\left( N \cdot \sum_{i=1}^{N} f_i \cdot P_i \right)}_{\mathcal{L}_{\text{aux}}}. \tag{2}$$

The auxiliary term encourages uniform utilization of the expert in a batch, where $f_i$ is the fraction of tokens routed to the expert $i$ and $P_i$ is its average routing probability.

## 2.2 THE SCALE SPARSE AUTOENCODER

### 2.2.1 MULTIPLE EXPERT ACTIVATION

Although Switch SAE is the first model to incorporate the MoE architecture into a sparse autoencoder, its implementation did not fully leverage the benefits of the paradigm. A key factor limiting both the interpretability and performance of Switch SAE is its low degree of expert specialization, resulting in significant feature redundancy across experts. This high redundancy explains why the performance of Switch SAE is often comparable to that of a standard TopK SAE (Figure 3).

This challenge is addressed by modifying the routing mechanism to transition from activating a single expert to activating $e$ experts, where $e \geq 2$ (Figure 1). The set of selected experts, denoted by $\mathcal{T}$, is determined by $\mathcal{T} = \text{argtopk}(\mathbf{x}, e)$. At the same time, the routing probabilities are computed using a softmax function: $\mathbf{p}(\mathbf{x}) = \text{softmax}(\mathbf{W}_{\text{router}}(\mathbf{x} - \mathbf{b}_{\text{router}}))$. For each selected expert $i \in \mathcal{T}$, initial feature activations are calculated as $f_i = \mathbf{W}_i^{\text{enc}}\mathbf{x}$. A crucial distinction in our approach is that the final sparse activations are determined through a global Top-K selection process across all activated experts. The sparse activation for the $j$-th neuron of the $i$-th expert is given by:

$$z_{ij} = f_{ij} \cdot \mathbb{I}\left(f_{ij} \in \text{TopK}\left(\{f_{l,m} \mid l \in \mathcal{T}\}\right)\right) \tag{3}$$

Subsequently, the output of each expert is calculated as $E_i(\mathbf{x}) = \mathbf{W}_i^{\text{dec}}\mathbf{z}_i$. The final reconstruction, $\hat{\mathbf{x}}$, is a weighted sum of these individual expert outputs:

$$\hat{\mathbf{x}} = \sum_i^{i \in \mathcal{T}} p_i(\mathbf{x}) E_i(\mathbf{x}). \tag{4}$$

The loss function for this model remains consistent with the one defined in Equation 2.

A significant innovation of this approach is the global activation mechanism, formulated in Equation 3. While its activation scope is more constrained than that of a dense SAE, it represents a fundamental departure from single-expert activation. This model can be viewed as an intermediate architecture between a traditional MoE SAE and a dense SAE, a concept explored further in Section 3.2.1.

### 2.2.2 FEATURE SCALING

A significant challenge in MoE SAE architectures is activation collapse, where diverse inputs activate a recurring, limited subset of experts and features. This phenomenon impairs reconstruction by limiting the information available to the decoder, thereby increasing both the MSE and feature redundancy. Our Feature Scaling mechanism is inspired by the principle of high-pass filtering from computer vision. The efficacy of such an approach for deep neural networks is supported by the work of Wang et al. (2022). In their research, they conclude that amplifying high-frequency features can counteract the "oversmoothing" phenomenon in Transformers. Adopting a similar philosophy, our method utilizes trainable parameters to amplify the high-frequency components of the encoder's features, enabling the encoded representations to preserve more fine-grained information. In Appendix H, we explain in detail how Feature Scaling is analogous to high-pass filtering.

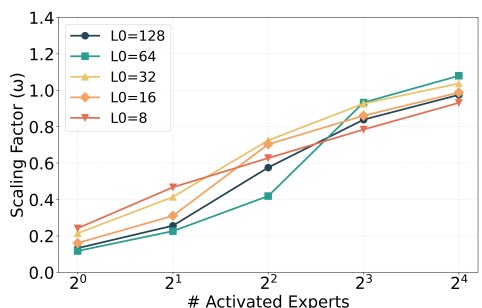

Figure 2: The scaling law for the trained scale factor $\omega$.

The Feature Scaling mechanism decomposes each expert's encoder weight matrix, $\mathbf{W}_i^{\mathrm{enc}}$. The implementation, illustrated in Figure 1, first defines a low-frequency component, $\bar{W}_i^{\mathrm{enc}}$, as the average feature vector of the expert's encoder weights: $\mathbf{w}_i = \frac{1}{n} \sum_{j=1}^{n} \mathbf{W}_{ij}^{\mathrm{enc}}$. Consequently, the high-frequency component, $\Delta\mathbf{W}_i^{\mathrm{enc}}$, is defined as the deviation from this low-frequency baseline: $\Delta\mathbf{W}_i^{\mathrm{enc}} = \mathbf{W}_i^{\mathrm{enc}} - \bar{W}_i^{\mathrm{enc}}$. The rationale for selecting this mean-based decomposition over alternative strategies is validated in Appendix E. It is crucial to distinguish our use of 'high-frequency' from its common use in SAE literature (i.e., 'frequently activating features'). Here, we use 'high-frequency' in its precise signal-processing sense. The mean vector $\bar{W}_i^{\mathrm{enc}}$ represents the shared 'DC component'. By subtracting this shared, low-frequency component, our mechanism effectively performs a high-pass filter. The remaining deviations $\Delta\mathbf{W}_i^{\mathrm{enc}}$, which capture the unique, distinguishing information for each feature, are thus precisely what we define as the 'high-frequency components'.

The final scaled weight matrix, $\hat{W}_i^{\mathrm{enc}}$, is constructed by amplifying this high-frequency component:

$$\hat{W}_i^{\mathrm{enc}} = \bar{W}_i^{\mathrm{enc}} + (1 + \omega)\Delta\mathbf{W}_i^{\mathrm{enc}} \tag{5}$$

The parameter $\omega$ is a trainable scaling factor that modulates the influence of the high-frequency component. Empirical analysis reveals two key properties of this parameter. First, it robustly converges to a positive value during training, indicating that the model learns to amplify high-frequency details, thereby enhancing its performance. Second, its magnitude exhibits a clear positive correlation with the number of activated experts, establishing a predictable scaling law (Figure 2).

## 3 RESULTS

We train sparse autoencoders with a hidden dimension of 768 on the intermediate activations of the 8th layer of the GPT-2 (Radford et al., 2019). The neurons in this layer contain rich semantic information beyond simple lexical features, making it an ideal testbed for assessing the monosemanticity of learned representations. The model is trained for 100,000 steps using the OpenWebText (Gokaslan et al., 2019). Additionally, we also trained on Gemma-2 2b, and the detailed analysis is in G.

## 3.1 END-TO-END PERFORMANCE

To ensure a fair comparison of computational efficiency, all models are evaluated under a strict FLOPS-matched paradigm. This approach normalizes the total number of floating-point operations across all models, allowing for a direct assessment of architectural benefits. We compare Scale SAE against three baseline models: Switch SAE (Mudide et al., 2025), Relu SAE (Bricken et al., 2023), TopK SAE (Gao et al., 2024), Gated SAE (Rajamanoharan et al., 2024a), and JumpReLU SAE (Rajamanoharan et al.). The hidden dimensions for each model are configured as follows to maintain this computational equivalence:

- **Scale SAE**: The total hidden dimension is set to 24,576. This is partitioned into settings of 256, 128, or 64 experts, with a corresponding activation of 8, 4, or 2 experts per forward pass.

- **Dense SAEs (TopK & Gated & ReLU & JumpReLU)**: The hidden dimension is set to 768. This architecture is strictly FLOPS-matched because the computational cost is determined by *active features*, not *total features*. While the MoE SAEs have a large 24,576-feature pool, they only activate 768 features per pass (e.g., Scale SAE with k=2 activates 2 experts $\times$ 384 features/expert = 768). The Dense SAE, by definition, activates all 768 of its features, resulting in an identical computational load for the forward pass.

- **Switch SAE**: This model is configured with a total hidden dimension of 24,576 and 32 experts, consistent with the Scale SAE's overall structure but limited to single-expert activation.

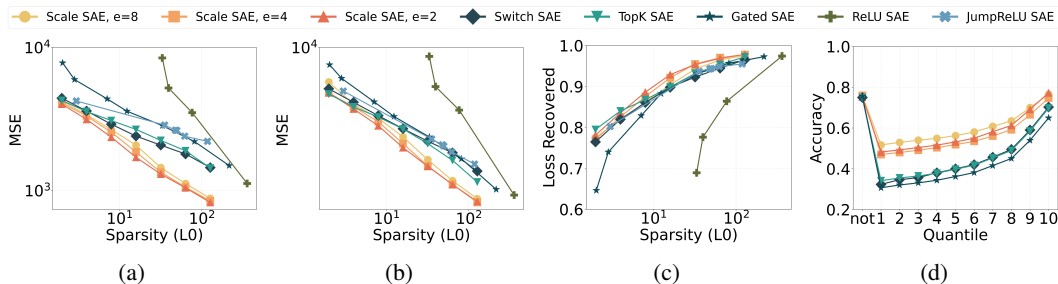

Figure 3: Performance comparison of Scale SAE against baseline models across three key metrics. (a, b) Reconstruction MSE on the OpenWebText and HLE-Biomedical datasets, respectively. (c) Loss Recovered on the HLE-Biomedical dataset. (d) Automated Interpretability Score on the OpenWebText2 dataset.

**Reconstruction MSE.** We benchmark the Reconstruction MSE of our model against several baselines across two distinct data domains. The first is the general-domain OpenWebText corpus, and the second is a specialized biomedical question-answering dataset, the Biomedical subset of Humanity's Last Exam (HLE-Biomedical) (Phan et al., 2025). On the OpenWebText dataset (Figure 3a), Scale SAE achieves a lower MSE than all baseline models without exception across all tested sparsity levels ($L_0$ norm). The performance advantage of Scale SAE becomes more pronounced as the target $L_0$ increases. At $L_0$ values of 32, 64, and 128, Scale SAE reduces the MSE by 37.21%, 41.99%, and 42.54%, respectively, compared to the next-best performing baseline. In contrast, the Switch SAE model fails to surpass the standard TopK SAE, despite possessing a 32-fold greater hidden dimension. This underperformance can be attributed to the high degree of feature redundancy among its experts. The results on the specialized HLE-Biomedical dataset (Figure 3b) further underscore the robustness of our approach. In this cross-domain experiment, the performance degradation of the Switch SAE model is even more pronounced, particularly at higher activation densities ($L_0 \geq 32$). In contrast, all three settings of Scale SAE maintain their superior performance, achieving the lowest reconstruction error in nearly all tested settings.

**Loss Recovered.** We next evaluate the Loss Recovered metric on the HLE-Biomedical dataset to assess model faithfulness as illustrated in the Figure 3c. The performance of the various multi-expert settings of Scale SAE ($e \in \{2, 4, 8\}$) is highly competitive and comparable to that of the TopK SAE baseline. Specifically, while TopK SAE holds a slight advantage in the low-sparsity regime ($L_0 \leq 4$), Scale SAE outperforms it as the number of activated features increases ($L_0 \geq 8$). The advantage of our model over other baselines is also evident, particularly when compared to Gated

SAE, which underperforms all other models, exhibiting the lowest Loss Recovered scores. Similarly, the Switch SAE model again fails to match the performance of the simpler TopK SAE, particularly at higher sparsity levels ($L_0 \geq 32$), which further suggests that its architecture suffers from significant feature redundancy. Although JumpReLU SAE performs very close to Scale SAE, its loss recovery is reduced by an average of 2.09% compared to Scale SAE. Furthermore, JumpReLU SAE performs far worse than Scale SAE on MSE. The substantial Loss Recovered scores of Scale SAE, combined with its low reconstruction MSE, validate that our proposed mechanisms achieve a superior balance between reconstruction fidelity and functional faithfulness.

**Automated Interpretability.** Interpretability is assessed using the automated scoring pipeline from Juang et al. (2024), following the methodology of Mudide et al. (2025). We test all models on the OpenWebText2 (Gao et al., 2020). Scale SAE's superior interpretability is evident across all activation quantiles, as detailed in Figure 3d. These quantiles (not, 1, 2, ...) represent the activation threshold required for a token to be associated with a feature. As expected, scores for all models improve at higher quantiles, as a stricter threshold isolates the most salient examples of a feature's activation. Crucially, Scale SAE consistently outperforms all baselines, including the structurally similar Switch SAE, at every quantile. Our proposed mechanisms enhance the monosemanticity and faithfulness of the learned features without compromising reconstruction accuracy or computational efficiency. In the Appendix C.3, we give a more detailed introduction to the Automated Interpretability pipeline.

## 3.2 ABLATION STUDY

### 3.2.1 MULTIPLE EXPERT ACTIVATION

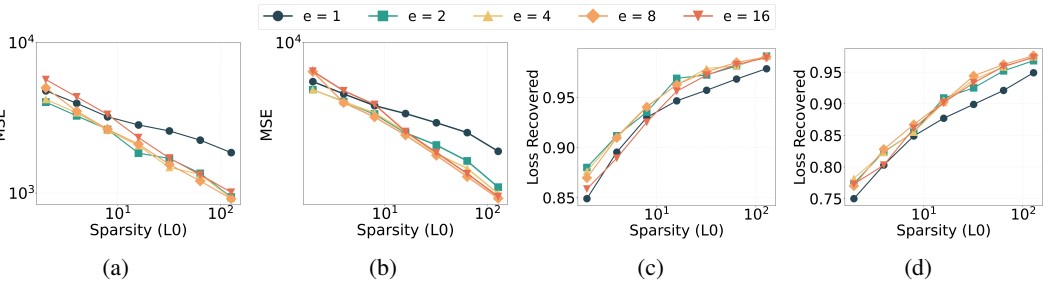

(a)        (b)        (c)        (d)

Figure 4: Performance comparison across two key metrics and distinct data domains, plotted as a function of the number of activated experts. (a, b) Reconstruction MSE was evaluated on the general-domain OpenWebText and the specialized HLE-Biomedical datasets, respectively. (c, d) Loss Recovered was evaluated on the same two datasets.

A key question is whether Scale SAE's performance improvements stem primarily from its multi-expert architecture. A focused ablation study addresses this by varying the number of activated experts and the feature sparsity level. We tested our model on the general-domain OpenWebText dataset. The results, depicted in Figures 4a and 4c, reveal a stark performance dichotomy. The single-expert setup ($e = 1$) always exhibits high reconstruction error and training instability. In contrast, all multi-expert setups ($e \geq 2$) achieve significantly improved accuracy and stability. Notably, the performance curves for these multi-expert models are tightly clustered, indicating that activating more than two experts yields diminishing returns. This principle is further underscored by the sharp decline in performance of the $e = 16$ models at low sparsity levels. To assess the generalizability of these findings, we replicate the experiment on the specialized HLE-Biomedical dataset (Figures 4b and 4d). The results were highly consistent with those on OpenWebText, confirming that the benefits of our Multiple Expert Activation strategy are also evident in a distinct, specialized domain.

### 3.2.2 FEATURE SCALING

While Multiple Expert Activation provides a robust architectural foundation, the second innovation, Feature Scaling, is designed to refine the learned representations. The specific contribution of this component was therefore isolated and quantified through a series of targeted experiments. The study directly compares the Reconstruction MSE and Loss Recovered for models with and without

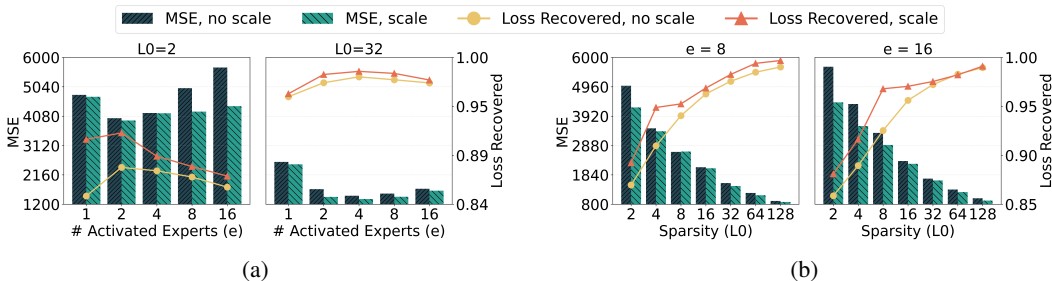

(a)                                                    (b)

Figure 5: The impact of Feature Scaling across different model settings. (a) Effect of Feature Scaling as a function of the number of activated experts, shown for fixed sparsity levels ($L_0 \in 2, 32$). (b) Effect of Feature Scaling as a function of target sparsity, shown for fixed expert setups ($e \in 8, 16$).

Feature Scaling across a range of expert and sparsity settings (Figures 5a and 5b). The analysis confirms the efficacy of Feature Scaling, with the mechanism leading to lower MSE and higher Loss Recovered across nearly all scenarios. Most strikingly, Feature Scaling resolves the primary failure mode identified in the previous analysis: the sharp performance decline of the $e = 16$ model at low sparsity levels. As shown in Figure 5b, this intervention almost entirely mitigates the instability, reducing MSE by an average of 10.9% for this setup while improving the Loss Recovered by 1.75%. The benefits also extend to the high-sparsity regime ($L_0 = 2$), where Feature Scaling improves Loss Recovered by an average of 4.132%. These results underscore that the Feature Scaling mechanism is not merely an incremental improvement but a critical component for ensuring training stability, particularly in variants with a high number of activated experts.

### 3.3 WHAT MAKES SCALE SAE WORK?

We hypothesize that the effectiveness of Scale SAE stems from two distinct mechanisms. The first hypothesis is that Multiple Expert Activation promotes a higher degree of expert specialization compared to single-expert routing. The second is that Feature Scaling enhances neuron activation diversity by amplifying the high-frequency components of the encoder's weights. This section provides a detailed analysis to validate these hypotheses, arguing that both mechanisms derive their benefits from a fundamental reduction in feature redundancy within the learned dictionary.

#### 3.3.1 EXPERT SPECIALIZATION

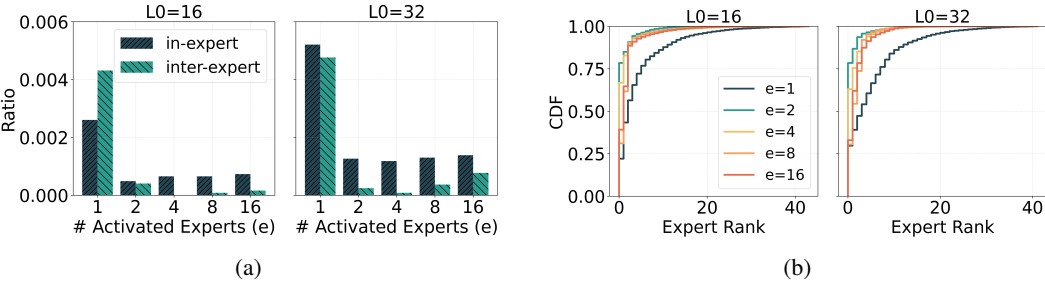

(a)                                                    (b)

Figure 6: Analysis of expert specialization and utilization under multi-expert settings. (a) Comparison of intra-expert versus inter-expert feature similarity across different numbers of activated experts. (b) Cumulative distribution of expert activation counts, with experts sorted by descending activation frequency.

The first hypothesis that Multiple Expert Activation promotes specialization can be validated by analyzing the geometric properties of the learned feature dictionaries. A primary metric for this analysis is feature redundancy, which is quantified as the proportion of features with a maximum cosine similarity to any other feature exceeding 0.9. High similarity in this context indicates significant semantic overlap (Braun et al., 2024).

Figure 6a reveals a clear distinction between the two architectures. Multi-expert models (e ≥ 2) demonstrate a strong and immediate signature of effective specialization compared to the single-expert baseline. They exhibit high intra-expert feature similarity, indicating strong internal coherence, while their inter-expert similarity remains low. Notably, all multi-expert configurations (e=2, 4, 8, 16) exhibit a similar level of inter-expert differentiation; we do not observe a clear trend of increasing specialization as the number of experts (e) increases beyond 2. Conversely, the single-expert model shows high similarity across all features, indicating significant feature redundancy and a failure to achieve structural specialization.

To further validate these findings, we investigate the distribution of expert activation using a specialized dataset of 1,500 past-tense verbs. The cumulative distribution function (CDF) of expert activation frequencies, plotted in Figure 6b, provides strong evidence of specialized learning. A steep CDF is a hallmark of specialization, as it indicates that a small subset of experts accounts for the majority of activations. The results are unambiguous: the single-expert model ($e = 1$) exhibits a flat CDF, confirming its inability to specialize for this targeted linguistic task. Conversely, **all multi-expert models exhibit much steeper CDFs, confirming that the mechanism effectively promotes specialized learning. However, as in the feature similarity analysis, the CDF slopes for $e = 4, 8, 16$ are highly similar to those for $e = 2$. The benefit of increased specialization shows clear signs of structural saturation, consistent with the optimal performance observed at lower levels of expert activation in our ablation study (Section 3.2.1). In Appendix B, we provide a mathematical proof that Multiple Expert Activation promotes specialization.

### 3.3.2 DIVERSITY OF NEURONAL ACTIVATION

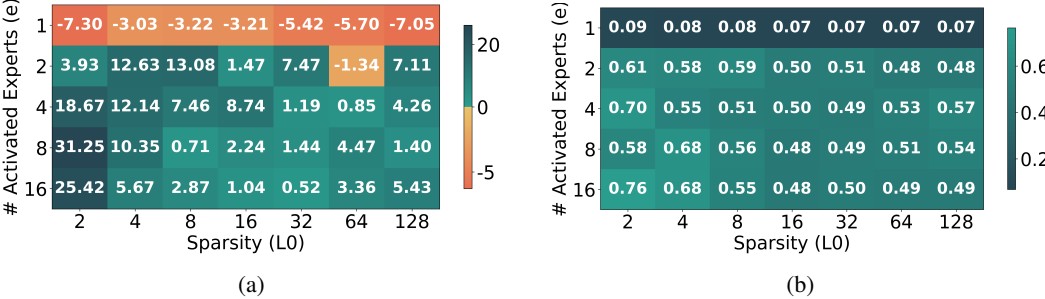

(a)  (b)

Figure 7: Impact of Feature Scaling on Neuron Activation Similarity. (a) Percentage reduction in activation similarity due to Feature Scaling. A positive value corresponds to a reduction in feature similarity. (b) Baseline neuron activation similarity (without Feature Scaling).

The mechanism by which Feature Scaling improves performance is elucidated through an analysis of its impact on the diversity of neuronal activation patterns. A curated set of 13,500 verb tokens, programmatically filtered from the OpenWebText, forms the basis for this analysis. We measure activation diversity by calculating the average pairwise similarity of activated neuron sets across these tokens. A higher average similarity score indicates a less diverse, more uniform activation pattern, signifying greater feature redundancy. A more formal description of this metric is given in the Appendix C.4.

Figure 7a shows a significant difference when using Feature Scaling. In the multi-expert activation setting, the average similarity of neuron activations decreases significantly as the scale increases. The average similarity decreases by 6.19%. It is worth noting that in the two cases of $L_0 = 2, e = 16$ and $L_0 = 2, e = 8$, the similarity decreases by 31.25% and 25.42%, respectively, which correspond precisely to the two cases shown in Section 3.2.2 where Feature Scaling has apparent optimization effects. However, in direct contrast to multi-expert settings, which leverage the mechanism to increase feature diversity (i.e., lower similarity scores), the single-expert ones exhibit an increase in feature similarity when scaling is applied. As shown in Figure 7b, the baseline single-expert model already exhibits a low feature similarity score. This apparent diversity, however, is misleading; it represents a state of high redundancy, where numerous features capture similar semantic information, leading to uninformative and chaotic activation patterns. Applying Feature Scaling to this

scenario does not simply reduce activation diversity; instead, it reduces feature redundancy, which remains helpful for improving interpretability. Appendix F provides more details on this experiment.

### 3.4 FEATURE SIMILARITY

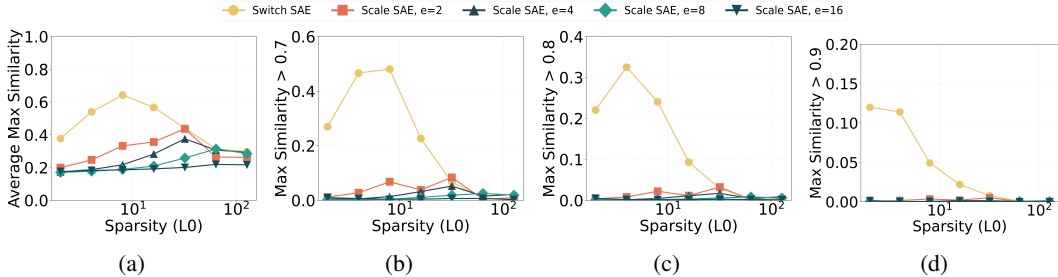

Figure 8: Analysis of Feature Similarity. (a) Average maximum cosine similarity. (b-d) Proportion of features with maximum similarity exceeding thresholds of 0.7, 0.8, and 0.9, respectively.

Prior work by Mudide et al. (2025) identifies feature redundancy as a primary factor limiting the performance of Switch SAE. To demonstrate that our proposed mechanisms directly address this issue, we employ a multi-faceted analysis of feature similarity. First, to evaluate redundancy, we analyze the average maximum cosine similarity (Figure 8a). We compare the Switch SAE against Scale SAEs with varying numbers of activated experts ($e \in \{2, 4, 8, 16\}$). As shown in the figure, at lower $L_0$ levels, the average similarity for Scale SAEs is significantly lower than that of the Switch SAE. When $L_0$ is large (e.g., 64 and 128), the values for both models tend to converge. Additionally, within the Scale SAEs, the similarity gradually decreases as the number of activated experts increases. Second, we examine the proportion of features with maximum similarity exceeding thresholds of 0.7, 0.8, and 0.9 (Figures 8b, 8c, and 8d). The trends across these three metrics are consistent. At lower $L_0$ levels, the Switch SAE exhibits a high proportion of redundant features, whereas Scale SAEs remain near zero across nearly all settings. The Switch SAE performance becomes comparable to that of Scale SAEs only when the target sparsity $L_0$ reaches 128.

### 3.5 COMPUTATIONAL EFFICIENCY

Table 1: Performance and efficiency comparison of Scale SAE against higher-budget TopK baselines.

| Model | Dict. Size | $L_0 = 32$ MSE ↓ | $L_0 = 32$ LR ↑ | $L_0 = 64$ MSE ↓ | $L_0 = 64$ LR ↑ | $L_0 = 128$ MSE ↓ | $L_0 = 128$ LR ↑ | FLOPs | Params | Mem. |
|---|---|---|---|---|---|---|---|---|---|---|
| **Scale SAE** | 1× | **1303.8** | 0.981 | **1044.6** | **0.990** | 825.8 | **0.993** | **7M** | **1.2M** | **4.7MB** |
| TopK SAE | 4× | 1565.9 | 0.980 | 1266.4 | 0.987 | 953.2 | 0.991 | 28M | 4.7M | 18.7MB |
| | 8× | 1329.1 | **0.985** | 1075.5 | **0.990** | **824.7** | **0.993** | 56M | 9.3M | 37.3MB |

Scale SAE is highly computationally efficient during both training and inference. To quantify this, we evaluated a Scale SAE model configured with 64 total experts, activating two experts per pass (totalling 7M FLOPs). We compared this against TopK SAE baselines with significantly higher computational budgets (4× and 8× FLOPs). We quantify efficiency based on the encoder's forward pass operations for FLOPs[1] and total parameter storage (assuming FP32 precision) for memory usage. The results (Table 1) demonstrate a superior performance-efficiency trade-off. Scale SAE not only outperforms the 4× TopK baseline but also achieves lower reconstruction error (MSE) than the 8× TopK model at $L_0 = 32$ and $64$, despite requiring only 12.5% of the FLOPs (7M vs. 56M). Under these metrics, Scale SAE demonstrates significant spatial advantages: it occupies merely 4.7MB of memory, compared to 37.3MB for the performance-equivalent 8× baseline. This confirms

---

[1]We estimate FLOPs as $6 \times P_{active} \times D$, where $P_{active}$ is the number of active parameters and $D$ is the token count. For TopK SAE, all parameters are active ($P_{active} = 2Md$). For Scale SAE, only selected experts are active ($P_{active} \approx 2Md/N_{total} \times N_{active}$).

that our approach achieves high interpretability without the prohibitive memory and compute costs typically associated with over-expanded dictionaries.

# 4 CONCLUSION

This paper fundamentally addresses the challenge of feature redundancy in the Mixture of Experts Sparse Autoencoder. We introduced Scale SAE, a framework that integrates two key insights to resolve the architectural and representational flaws of prior models. First, and most significantly, we challenge the prevailing single-expert activation paradigm that has limited prior MoE-SAE work. We demonstrate that this rigid $e = 1$ assumption was the actual underlying bottleneck, and that the widely-cited 'feature redundancy' was merely its symptom. Our finding is that a minimal shift to co-activating at least two experts is sufficient to unlock massive performance gains. Building on this insight, our second mechanism, Feature Scaling, provides a learnable method to further enhance feature diversity by amplifying high-frequency components. By resolving these core issues, our work demonstrates that the MoE SAE paradigm is a promising approach for reconciling mechanistic interpretability with computational efficiency in LLM analysis.

**Limitations.** Despite its strong performance, Scale SAE presents several avenues for future research. (1) A primary direction is to address the performance gap that persists between Scale SAE and dense models under width-matched conditions, potentially by exploring advanced routing algorithms, such as the C2R strategy (Zhang et al., 2025). (2) Furthermore, a key challenge remains in deepening expert specialization to achieve a more fine-grained conceptual decomposition; the persistence of expert-level redundancy continues to limit direct interpretability (Appendix D), making resolution a crucial next step. (3) Finally, our analysis of Multiple Expert Activation focuses solely on enhancing expert specialization; we must investigate other underlying factors and mechanisms, as understanding these deeper causes will be crucial for creating more advanced MoE SAE architectures.

# 5 RELATED WORK

**Signal Processing on Deep Learning Architectures.** Frequency-domain analysis, inspired by classical signal processing (Daubechies, 1992; Oppenheim & Schafer, 2009), is instrumental in addressing limitations in deep learning. For instance, Lee-Thorp et al. (2022) replace self-attention with the Fourier Transform to improve computational efficiency, while others have used similar techniques to capture long-range dependencies (Rao et al., 2021) or enable efficient super-resolution (Xie et al., 2021). Critically, this analytical approach is also used to correct architectural flaws. Wang et al. (2022) identify the self-attention mechanism as an intrinsic low-pass filter and proposed amplifying high-frequency features to counteract the resulting "oversmoothing" phenomenon.

**Sparse Autoencoders.** SAEs decompose polysemantic activations into an overcomplete feature dictionary; this approach is pioneered for LLMs by Anthropic(Cunningham et al., 2023; Bricken et al., 2023). Subsequent work introduces a series of architectural and training modifications to the original SAE framework. To enforce sparsity more directly, Gao et al. (2024) propose the TopK SAE, which selects only the strongest $K$ features per token. Another line of work focuses on the activation function itself, with Taggart (2024) and Rajamanoharan et al. (2024b) introducing learnable, threshold-based activations that replace the standard ReLU. To mitigate bias in feature magnitude estimation, Rajamanoharan et al. (2024a) develop the Gated SAE, which decouples feature selection from its magnitude by applying the sparsity penalty only to a dedicated selection gate.

**Mixture of Experts Models.** The core MoE concept of conditional computation is first proposed by Jacobs et al. (1991). Over two decades later, Shazeer et al. (2017) are the first to successfully apply the paradigm to large-scale deep learning. A series of subsequent studies further push the computational efficiency of MoE to its limits through innovations such as introducing the auxiliary load-balancing loss for stable training (Lepikhin et al., 2020), simplifying the routing mechanism to a single expert (Fedus et al., 2022), and successfully scaling these architectures to over a trillion parameters (Du et al., 2022). The versatility of this approach has since been demonstrated by its expansion into the computer vision domain (Riquelme et al., 2021) and its adoption in state-of-the-art language models such as Mixtral 8x7B (Jiang et al., 2024).

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

CONTENTS OF THE APPENDIX

## A USAGE OF LLM

We employed Google's Gemini 2.5 Pro and OpenAI's GPT-5 as writing assistance tools during the preparation of this manuscript. Their role was exclusively for language refinement, such as improving readability and rephrasing for clarity in an academic writing style. This usage aligns with standard academic practices for language polishing.

## B PROOF

**Setup.** Let $x \in \mathbb{R}^d$ be the input. The model consists of $M$ experts $\{E_m\}_{m=1}^M$. A router selects a subset $\mathcal{K}$ of size $k$ with binary activation $g_m \in \{0, 1\}$. The expert output aggregates internal features: $E_m = \sum_{f_i \in \Phi_m} a_i f_i$ ($\|f_i\| = 1$). The loss is $\mathcal{L} = \frac{1}{2} \mathbb{E}_x[\|x - \sum_m g_m E_m\|^2]$.

### B.1 PROPOSITION 1: INCREASED EXPERT SPECIALIZATION

*Proof.* We analyze the fixed-point solution $E_m^*$ derived from the stationarity condition $\nabla_{E_m} \mathcal{L} = 0$. The general solution is given by:

$$E_m^* = \frac{\mathbb{E}_x \left[ g_m \left( x - \sum_{n \neq m} g_n E_n \right) \right]}{\mathbb{E}_x[g_m]} \tag{6}$$

**Case $k = 1$.** The mutual exclusivity constraint implies $g_m = 1 \implies g_{n \neq m} = 0$. Consequently, the interaction term $\sum_{n \neq m} g_n E_n$ vanishes identically. The fixed point degenerates to:

$$E_m^* = \mathbb{E}_x[x \mid g_m = 1] \tag{7}$$

The expert $E_m$ convergesfeature indices centroid of the raw data, necessitating the encoding of composite concepts (Generalist behavior).

**Case $k > 1$.** The co-activation allows $g_n = 1$ for $n \neq m$. The interaction term is non-zero, representing the approximation by other experts $\hat{x}_{\neg m}$. The fixed point becomes:

$$E_m^* = \mathbb{E}_x[x - \hat{x}_{\neg m} \mid g_m = 1] \tag{8}$$

The expert $E_m$ converges to the centroid of the residual. This forces $E_m$ to encode the subspace orthogonal to $\hat{x}_{\neg m}$, isolating unique atomic features (Specialist behavior).

### B.2 PROPOSITION 2: REDUCED FEATURE REDUNDANCY ($k > 1$)

*Proof.* Let $\rho_{ij} = \langle f_i, f_j \rangle$ be the cosine similarity between feature $f_i \in E_m$ and $f_j \in E_n$ ($m \neq n$). We analyze the gradient of $\mathcal{L}$ with respect to $\rho_{ij}$, derived from the second-order expansion of the loss:

$$\frac{\partial \mathcal{L}}{\partial \rho_{ij}} \propto \mathbb{E}_x \left[ (g_m a_i)(g_n a_j) \right] \tag{9}$$

**Case $k = 1$.** The exclusivity constraint enforces $g_m g_n \equiv 0$ for all $m \neq n$.

$$\frac{\partial \mathcal{L}}{\partial \rho_{ij}} = 0 \tag{10}$$

The gradient field implies no repulsive force between features in different experts. Experts independently fitting overlapping data manifolds leads to $\rho_{ij} \to 1$ (High Redundancy).

**Case $k > 1$.** The co-activation probability $P(g_m = 1, g_n = 1) > 0$. For correlated active features ($a_i, a_j > 0$):

$$\frac{\partial \mathcal{L}}{\partial \rho_{ij}} > 0 \tag{11}$$

The loss landscape imposes a strictly positive penalty on feature alignment. Gradient descent minimizes this potential by driving $\rho_{ij} \to 0$ (Orthogonalization). $\qed$

## C  DETAILED EVALUATION METRICS

The performance of each Sparse Autoencoder (SAE) variant was assessed using a suite of three key metrics, each designed to evaluate a different aspect of the model's quality: reconstruction fidelity, faithfulness to the original model, and the interpretability of its learned features.

### C.1  RECONSTRUCTION MEAN SQUARED ERROR (MSE)

This metric quantifies the fidelity of the SAE's reconstruction by measuring the average squared difference between the original and reconstructed activation vectors. A lower MSE signifies a more accurate reconstruction and less information loss. It is formally defined as:

$$\mathcal{L}_{\text{MSE}} = \frac{1}{D} \sum_{i=1}^{D} (x_i - \hat{x}_i)^2 \tag{12}$$

where $x \in \mathbb{R}^D$ is the original activation vector from the Large Language Model (LLM), and $\hat{x}$ is the corresponding vector reconstructed by the SAE. However, a low MSE alone is an insufficient measure of quality, as it can be achieved by learning polysemantic features. It must therefore be assessed in conjunction with the other metrics.

### C.2  LOSS RECOVERED

This metric assesses the SAE's *faithfulness* to the original model by measuring the extent to which its sparse features can account for the LLM's downstream performance. A higher Loss Recovered score shows that the SAE has successfully captured the features most salient to the LLM's predictions. The calculation involves three forward passes through the LLM on a given dataset:

- First, the standard cross-entropy loss is calculated using the original, unmodified activations, yielding $\mathcal{L}_{\text{original}}$.
- Second, the loss is recalculated with the SAE's reconstructed activations substituted at the target layer, yielding $\mathcal{L}_{\text{reconstructed}}$.
- Third, a baseline loss is established by zeroing out the activations at the target layer, yielding $\mathcal{L}_{\text{zero}}$.

The Loss Recovered score is the fraction of the performance drop caused by this zero-ablation that is "recovered" by the SAE's reconstruction, formally defined as:

$$\text{Loss Recovered} = \frac{\mathcal{L}_{\text{zero}} - \mathcal{L}_{\text{reconstructed}}}{\mathcal{L}_{\text{zero}} - \mathcal{L}_{\text{original}}} \tag{13}$$

### C.3  AUTOMATED INTERPRETABILITY SCORE

To objectively quantify the *monosemanticity* of individual features, we employ the Automated Interpretability method, a two-stage pipeline first proposed by Juang et al. (2024). This approach automatically assesses the conceptual consistency of the text passages that trigger a given feature. The pipeline consists of the following two stages:

- **Stage 1: Max Activation Curation.** The MaxAct (Bricken et al., 2023) identifies a corpus of text passages from a large dataset that cause a feature to activate most strongly. These passages are assumed to represent the feature's core semantic meaning.
- **Stage 2: Automated Scoring.** An external, powerful LLM (Llama-3 in this study) is prompted to act as an automated judge. It is presented with a held-out text passage that also activates the feature and performs a classification task: to determine if the new passage is conceptually consistent with the core meaning established by the MaxAct corpus.

The final interpretability score for a feature is the LLM's classification accuracy on this task. A high score signifies that the feature represents a single, coherent concept, which is the definition of high monosemanticity.

Table 2: The top-8 features with the highest activation intensity and their corresponding semantic explanations.

| Expert ID | Feature ID | Activation Strength | Most Relevant Interpretation |
|---|---|---|---|
| 0 | 288 | 17.1770 | plural nouns |
| 0 | 121 | 15.5380 | plants |
| 0 | 86 | 14.4454 | countable nouns |
| 50 | 46 | 13.7324 | Apple/ios |
| 0 | 232 | 13.6768 | fruits |
| 53 | 9 | 12.9206 | Polysemantic Feature |
| 50 | 98 | 11.2872 | Apple/ios |
| 61 | 84 | 11.0593 | Ending with s |

### C.4 AVERAGE NEURONAL ACTIVATION SIMILARITY

This metric is designed to provide a single, quantitative measure of the overall feature redundancy within the learned SAE dictionary. It is calculated by iterating through every unique pair of tokens in a dataset, computing the proportion of shared activated neurons for each pair, and then averaging these scores across all pairs. It is formally defined as:

$$\text{Similarity} = \frac{1}{N(N-1)} \sum_{i \neq j} \frac{k_{i,j}}{K_{\text{total}}} \tag{14}$$

where $N$ is the total number of tokens, $k_{i,j}$ is the number of common neurons activated by both token $i$ and token $j$ (i.e., $|S_i \cap S_j|$), and $K_{\text{total}}$ is a normalization factor representing the total number of neurons activated per token (e.g., the value of $K$ in a TopK SAE).

The final score provides a global measure of the dictionary's representational efficiency. A high score (closer to 1) indicates a high degree of *feature redundancy*, suggesting that different tokens activate very similar sets of neurons and that many features capture overlapping semantic concepts. Conversely, a low score (closer to 0) indicates a high degree of *feature diversity*, suggesting that individual features have learned to represent more specific, monosemantic concepts. Therefore, this metric serves as a crucial tool for quantifying the ability of different SAE architectures to learn a well-decomposed feature space.

## D CASE STUDY: INTERPRETING THE TOKEN "APPLES"

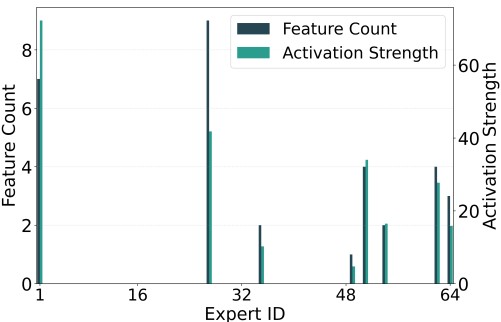
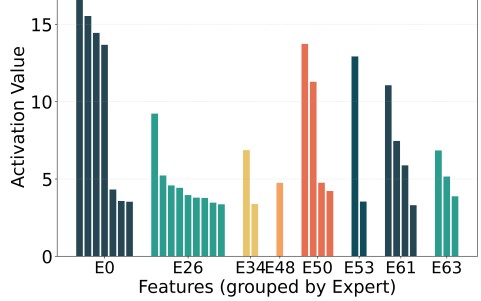

(a) The intensity of different experts being activated.

(b) The intensity of activation of different features (grouped by expert).

Figure 9: The token "apples" in "I love **apples** but hate oranges completely." is interpreted by Scale SAE at the 8th layer of GPT2.

To qualitatively assess the interpretability of Scale SAE, we analyzed the feature activations for the token "apples" in the sentence "I love **apples** but hate oranges completely." For each feature, its

semantic meaning was derived by manually summarizing the concept familiar to the top five tokens that maximally activated it. Features for which a unified semantic concept could not be determined were labeled as "Polysemantic."

### D.1 Multi-Faceted Interpretation

The Scale SAE router selected eight experts to represent the token, with the top 32 most active features drawn from across this expert subset (Figure 9). This global selection, involving multiple experts rather than a single one, is a key architectural advantage. An analysis of the most salient features (Table 2) reveals a multi-faceted decomposition of the token's meaning. The model successfully identified several distinct conceptual layers:

- **Lexical and Syntactic Features:** Expert 0 contributed features corresponding to "plural nouns" (feature 228) and "countable nouns" (feature 86).
- **Semantic Features:** Other experts captured core semantic concepts related to "apples," such as "plants" and "fruits."
- **Orthographic Features:** The model also identified structural patterns, such as expert 61's feature 84, which responds to tokens ending in "s."

Interestingly, the model also activated features in expert 50 related to the proper noun "Apple" (the company), demonstrating its ability to capture even contextually incorrect but related concepts.

### D.2 Discussion and Limitations

Despite these positive results, this detailed analysis also highlights several limitations and avenues for future improvement. First, some polysemantic features persist. While Scale SAE significantly improves monosemanticity over baseline models, some features—such as feature 9 in expert 53—are still activated by a broad and diverse range of tokens, rendering their specific semantic interpretation difficult. Second, intra-expert semantic redundancy was not eliminated; for example, two distinct features in expert 50 were found to represent the same concept. Third, the model's reliance on highly abstract grammatical features (e.g., "plural noun") suggests a lack of fine-grained semantic decomposition. Finally, expert specialization remains imperfect, as exemplified by expert zero activating features for disparate conceptual categories (lexical, syntactic, and semantic), which contradicts the goal of having each expert focus on a distinct domain.

## E Further Study on Feature Decomposition Methods

Our Feature Scaling mechanism is predicated on the decomposition of an expert's encoder weight matrix, $M$, into a "low-frequency" component ($M_{lp}$) and a "high-frequency" component ($M_{hp}$), such that $\hat{M} = (M - M_{lp}) \times \text{Scale} + M_{lp}$. The primary implementation in this work defines the low-frequency component as the mean of the feature vectors, $M_{lp} = M_{\text{mean}}$. To validate this design choice, we conducted an ablation study comparing our approach against two alternative decomposition strategies.

### E.1 Alternative Decomposition Strategies

We evaluated the following three methods for defining the low-frequency and high-frequency components:

- **Method 0 (Our Proposed Method): Mean-based Decomposition.** The low-frequency component is the average feature representation of the expert. This method hypothesizes that the mean captures the most common, low-frequency patterns, while deviations from the mean represent unique, high-frequency details.

$$\hat{M} = (M - M_{\text{mean}}) \times \text{Scale} + M_{\text{mean}} \tag{15}$$

- **Method 1: Identity-based Decomposition.** The low-frequency component is defined as the identity matrix, $I$. This strategy frames the problem as scaling the transformative part

of the weight matrix $(M - I)$ relative to the identity-preserving part $(I)$.

$$\hat{M} = (M - I) \times \text{Scale} + I \tag{16}$$

- **Method 2: Learning-based Decomposition.** The low-frequency component is a learnable matrix, $A_{LP}$, which is trained concurrently with the rest of the network. This method allows the model to dynamically learn the optimal separation of low- and high-frequency components.

$$\hat{M} = (M - A_{LP}) \times \text{Scale} + A_{LP} \tag{17}$$

## E.2 DISCUSSION OF RESULTS

We compared the performance of these three decomposition strategies across a range of sparsity levels, with the results visualized in Figure 10. The data reveal that Method 1 (Identity-based) always results in a significantly higher reconstruction error, establishing it as a sub-optimal approach. Conversely, our proposed Method 0 (Mean-based) and the more complex Method 2 (Learned-based) both achieve strong and highly competitive reconstruction performance. As the performance difference between these two methods is marginal and often within the range of typical training variance, there is no compelling evidence to justify the added complexity of the learning-based approach. The learning-based method introduces additional trainable parameters without offering a significant performance benefit over our

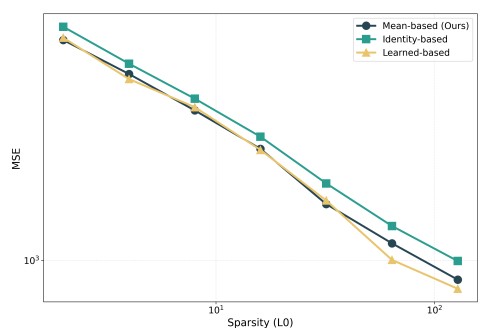

Figure 10: Performance comparison of the three feature decomposition strategies across a range of sparsity levels.

simpler, more efficient mean-based strategy. Therefore, given that the primary goal is to reconstruct activations faithfully, the superior performance-to-complexity ratio of our mean-based decomposition validates it as the most effective and well-balanced strategy.

## F DISTRIBUTION OF NEURON ACTIVATION

To provide a more granular analysis of feature similarity, this section supplements Section 3.3.2 by examining the distribution of the number of overlapping features, $k_{i,j}$, between token pairs. The experimental setup remains consistent, with a total of 64 experts, a varying number of activated experts ($e \in \{1, 2, 4, 8, 16\}$), and varying sparsity ($L_0 \in \{2, 4, \ldots, 128\}$).

The distributions of $k_{i,j}$ for models with and without Feature Scaling are presented in Figure 11. A consistent bimodal pattern was observed across all scenarios. The primary mode, representing the highest proportion of token pairs, is centred at zero overlap ($k_{i,j} = 0$). A smaller, secondary mode represents pairs with non-zero overlap. We identified several key findings from these distributions.

First, in the single-expert settings, the distribution is heavily dominated by the zero-overlap mode. This seemingly high diversity is, in fact, an artefact of extreme feature redundancy; semantically similar tokens often activate entirely different sets of redundant features, resulting in low pairwise overlap. Second, in multi-expert settings, the secondary mode becomes more prominent, particularly as activation density increases ($L_0 > 16$). This indicates that these models learn to represent similar tokens using shared, overlapping feature sets, which is a hallmark of effective specialisation. Finally, applying Feature Scaling consistently shifts this secondary mode to the right. This shift demonstrates that, for semantically related token pairs, our mechanism encourages the encoder to activate more diverse, less overlapping combinations of neurons, thereby further enhancing feature diversity.

## G TRAINING ON ADDITIONAL MODEL

Figure 12 illustrates the comprehensive performance evaluation of our proposed Scale SAE compared to the Switch SAE baseline on the Gemma-2 2b model (Team et al., 2024). We report results

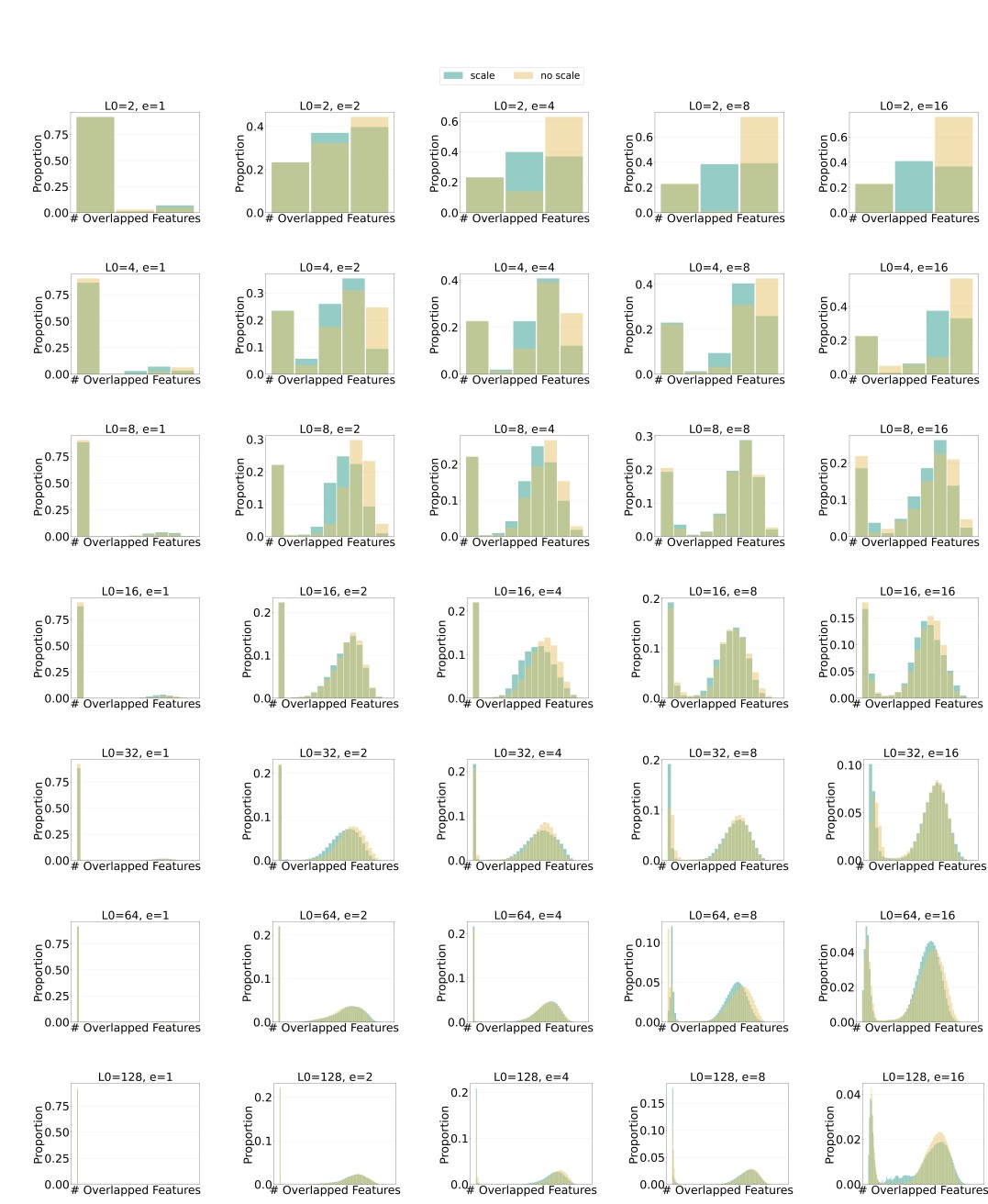

Figure 11: Distributions of the number of overlapping features for models with and without Feature Scaling across various expert activation and sparsity ($L_0$) models.

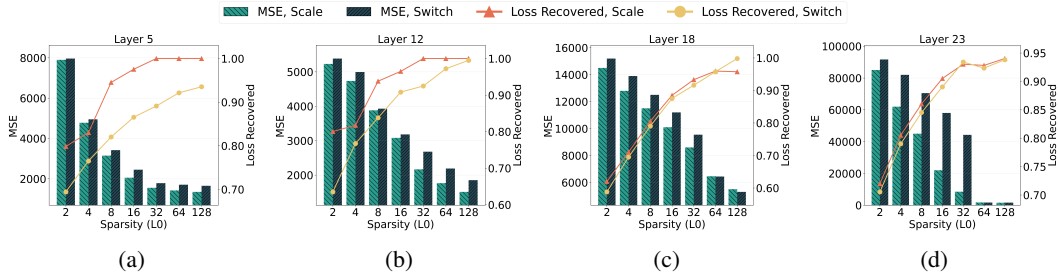

Figure 12: Performance comparison on Gemma-2 2b across Layers 5, 12, 18, and 23. Each subplot illustrates both **Reconstruction MSE** and **Loss Recovered** as a function of the number of active features ($L_0$). The Scale SAE consistently achieves lower reconstruction error and faster loss recovery saturation compared to the Switch SAE baseline across varying network depths.

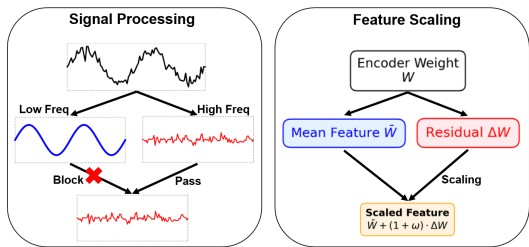

Figure 13: Schematic analogy between high-pass filtering and Feature Scaling.

across four distinct layers (5, 12, 18, and 23) to assess performance at different network depths. For a fair comparison, both architectures maintain a fixed dictionary width of $24,576$. The Scale SAE is configured with 64 experts using top-2 routing, while the Switch SAE utilises 32 experts. All evaluations are conducted on the OpenWebText dataset.

**Reconstruction MSE.** The Scale SAE consistently achieves lower Reconstruction MSE across all tested layers and sparsity levels ($L_0$). In the earlier and middle layers (Layers 5 and 12), the Scale SAE demonstrates a clear advantage. For instance, in Layer 5 at $L_0 = 32$, the Scale SAE reduces the MSE to $1550.45$, significantly outperforming the Switch SAE's $1781.96$. This trend persists in deeper layers; in Layer 23, where reconstruction is inherently more challenging due to feature abstraction, the Scale SAE maintains superior fidelity, avoiding the sharp "cliffs" in error rates observed in the baseline at lower $L_0$ values.

**Loss Recovery Efficiency.** The advantage of the Scale SAE is most pronounced in the Loss Recovered metric, where it demonstrates significantly higher efficiency per active feature. In Layers 5 and 12, the Scale SAE approaches perfect loss recovery (Loss Recovered $\approx 1.0$) much faster than the baseline. Specifically, in Layer 12, the Scale SAE achieves full recovery ($1.0$) at $L_0 = 32$, whereas the Switch SAE only reaches $0.92$ at the same sparsity level and requires $L_0 = 128$ to approach saturation. In Layer 18, the Scale SAE achieves a Loss Recovered score of over $0.99$ by $L_0 = 64$, while the Switch SAE lags at $0.95$. Even in Layer 23, the Scale SAE exhibits a smoother and more monotonic recovery curve. Overall, these results suggest that the multiplicative scaling mechanism in Scale SAE allows for more expressive feature composition, enabling it to capture essential model behaviours with fewer active latents compared to the additive nature of Switch SAE.

## H ANALOGY TO SIGNAL PROCESSING

To provide intuition for our method, we draw an analogy between Feature Scaling and High-Frequency Emphasis in signal processing (Figure 13). In signal analysis, a raw signal comprises a low-frequency baseline (coarse trend) and high-frequency fluctuations (fine details); high-pass filters isolate these details by attenuating the baseline. Similarly, within our framework, we view the "Mean Feature" of an expert cluster as the low-frequency trend representing shared redundancy,

while the "Residual Features" act as the high-frequency details representing unique, specialized information. Our Feature Scaling mechanism introduces a learnable scaling factor $\omega$ applied to the residual component $\Delta \mathbf{W}$. While this formulation mathematically allows for flexible scaling, our experiments demonstrate that the model consistently learns to amplify these residuals (i.e., $\omega > 0$), effectively functioning as a high-frequency emphasis filter. This learned amplification increases the relative importance of fine-grained differences over the shared mean, forcing the model to distinguish features based on their specific deviations rather than redundant commonalities.

