# OpenReview forum: "Beyond Redundancy: Diverse and Specialized Multi-Expert Sparse Autoencoder"
_ICLR.cc/2026/Conference — Submitted to ICLR 2026_

### Official Review · Reviewer_S4XQ · 2025-10-15

**Soundness:** 2
**Presentation:** 2
**Contribution:** 3
**Rating:** 2
**Confidence:** 3

**Summary:**

The paper introduces multi expert SAEs (as opposed to single expert SwitchSAEs). They further introduce feature scaling as a mean of encouraging specialisation and reducing feature redundancy. They run evaluations on gpt2 on fraction of loss recovered, mse, interpretability, and redundancy of features. They show improvements again some competing architectures of SAEs.

**Strengths:**

The introduction of weighted MoE SAEs is novel, and feature scaling is effective as a way to enhance specialisation (which could be tied to interpretability, as an interesting future direction).
There are many evaluations criteria, and improvements are shown across them.
The problem of SAE feature interpretability, and the improvement of SAEs are well located problems within the literature.

**Weaknesses:**

The main limitations are:
- evaluations restricted to gpt2
- the improvements on L0 against MSE/loss look incremental
- only.3 architectures (topk/switch/gated) are tested against (more baselines could help, such as matryoshka or jumprelu saes)

I would increase my score if evaluations were successfully (more than incremental improvements) run on more models/families.

Style:
- figure 3 looks cluttered (make all your saes one color and all the baselines another)

**Questions:**

Have you tried expanding evaluations to other models or families (even small ones like pythia 70m)?

Is fraction of loss recovered the LLM's loss or the SAE's EV?

Have you tried experiments linking the specialisation of features to their interpretability?

---

> ### Author Response · Authors · 2025-11-18
> **Response to Reviewer S4XQ (Part 1)**
>
> We sincerely thank the reviewer S4XQ for the thoughtful evaluation and constructive feedback. We are deeply encouraged by the reviewer's upbeat assessment and are especially grateful for their recognition of our paper's key strengths. We are pleased that the reviewer valued the novelty of our weighted MoE SAEs, the effectiveness of feature scaling for specialization, our comprehensive evaluation criteria that show broad improvements, and our work's focus on well-located problems within the literature. This supportive feedback is very motivating. We address the key concerns and suggestions below:
>
>
> **Weaknesses: 1:** Evaluations restricted to gpt2
>
>
> **Response:** Thank you for raising this concern. We are actively working on these experiments(On Gemma-2 2b). In Appendix F, Figure 12 shows the performance of training on layer 12 of the Gemma-2 2b. The Scale SAE achieves lower reconstruction MSE across all sparsity levels, especially as the density increases gradually. The advantage of the Scale SAE in Loss Recovered is even more pronounced, showing a steeper recovery rate of the original layer's loss and better fidelity across all tested sparsity levels.
>
>
> **Weaknesses: 2:** The improvements on L0 with respect to MSE/loss look incremental.
>
> **Response:** Thank you for raising this concern. However, we disagree with the characterization of our results as "incremental." The core challenge in SAEs is the fundamental trade-off between L0 sparsity and reconstruction quality (MSE / Loss Recovered). As clearly demonstrated in Figure 3, our Scale SAE achieves a Pareto Improvement over the Switch SAE baseline. This defines our primary quantitative contribution: at any given L0 sparsity level, our model simultaneously achieves a significantly lower MSE and a substantially higher Loss Recovered. Fully improving the SOTA's trade-off boundary in this manner is a substantial, not incremental, advancement in the field.
>
>
> Furthermore, our contribution is not limited to performance numbers. As stated in our introduction (Section 1), a primary, non-incremental contribution is the introduction and deep mechanistic analysis of why our method works. We introduce two synergistic mechanisms (Multiple Expert Activation and Feature Scaling) and provide a detailed analysis (Section 4) that establishes how Multiple Expert Activation improves specialization and how Feature Scaling reduces feature redundancy. This mechanistic insight into why and how our model succeeds is, in itself, a significant scientific contribution to the field.
>
> **Weaknesses: 3:** Only three architectures (topk/switch/gated) are tested against (more baselines could help, such as matryoshka or jumprelu saes)
>
>
> **Response:** Thank you for this suggestion. Our experimental design aligned with the prior state-of-the-art, Switch SAE [1], which focused on comparisons with other relevant MoE-SAE variants, such as Gated SAE and TopK SAE. To further strengthen our analysis and provide a more comprehensive comparison, we have now also included a standard ReLU SAE in our baseline suite, with the results added to the revised paper (Figure 1).
>
>
> [1] Mudide, A., Engels, J., Michaud, E. J., Tegmark, M., & De Witt, C. S. (2024). Efficient Dictionary Learning with Switch Sparse Autoencoders. ArXiv. https://arxiv.org/abs/2410.08201
>
> **Weaknesses: 4:** Style: Figure 3 looks cluttered (make all your saes one color and all the baselines another)
>
> **Response:** Thank you for your suggestion; I will fix it in the revised version.

---

> ### Author Response · Authors · 2025-11-18
> **Response to Reviewer S4XQ (Part 2)**
>
> **Question 1:** Have you tried expanding evaluations to other models or families (even small ones like Pythia 70m)?
>
>
> **Response:** We are actively working on these experiments (Gemma-2 2b). Should these results become available before the end of the discussion period, we will post an update here immediately. We fully commit to including these results in the appendix of the final version of the paper.
>
>
> **Question 2:** Is the fraction of loss recovered the LLM's loss or the SAE's EV?
>
>
> **Response:** Thanks for your question. We confirm that 'Loss Recovered' refers to the LLM's downstream cross-entropy loss, not the SAE's EV (which typically measures L2 reconstruction fidelity). Our metric specifically measures the functional faithfulness of the SAE's features to the LLM's predictions. We define this metric and its calculation explicitly in Appendix B.2.
>
>
> **Question 3:** Have you tried experiments linking the specialisation of features to their interpretability?
>
> **Response:** That is an insightful question. Our analysis in the paper has focused on the expert-level specialization rather than linking individual features to their interpretability.
>
>
> As we demonstrate in Section 3.3.1, our Scale SAE experts exhibit significantly stronger specialization (i.e., they fire on more distinct, coherent concepts) compared to the baseline.
>
>
> While we have not proposed a formal quantitative metric for this specialization, our core argument is built upon the direct comparison between the models:
>
>
> The Switch SAE baseline suffers from low expert specialization (as shown in 3.3.1) and achieves low interpretability.
>
>
> Our Scale SAE achieves high expert specialization (3.3.1) and achieves high interpretability.
>
>
> This comparative evidence strongly supports our hypothesis that enhancing expert-level specialization is a key mechanism for achieving higher interpretability.

---

> ### Comment · Reviewer_S4XQ · 2025-11-19
>
> Dear authors,
>
> Thanks for your rebuttal,
>
> I will increase my score due to the extra experiments, but I remain convinced that it would help to add:
> - all layers of gemma
> - more baselines (relu isn't a strong one)
> - fix figure 3

---

> > ### Author Response · Authors · 2025-11-21
> > **response to reviewer S4XQ**
> >
> > Dear reviewer,
> >
> > Thank you for your recognition. Below is my response to the new comment.
> >
> > Weakness 1: More baselines
> >
> > Response: We trained a JumpRelu SAE and compared it with Scale SAEs on the first three metrics (MSE on the OpenWebText and HLE-Biomedica, and Loss Recovered on the HLE-Biomedical) (Figure 3).
> >
> > Although JumpRelu SAE's Loss Recovered performance is very close to Scale SAE, it is on average 2.09% lower. Furthermore, JumpRelu SAE's performance on MSE is significantly worse than Scale SAE.
> >
> > Weakness 2: Fix Figure 3
> >
> > Response: We modified Scale SAE; although not completely identical, they all use orange tones, which we believe contributes to visual distinctiveness.
> >
> > Weakness 3: All layers of gemma
> >
> > Response: Although we believe the experiments on layer 12 have demonstrated the efficiency of Scale SAE, we will also run the same experiments on layers 8, 18, and 23 and report the results as soon as possible. Gemma has 26 layers, making it virtually impossible to run experiments across all layers.

---

> > > ### Author Response · Authors · 2025-11-25
> > > **response to reviewer S4XQ**
> > >
> > > Dear reviewer,
> > >
> > > As we mentioned in our previous response, we trained on layers 5, 12, 18, and 23 of Gemma-2 2b and supplemented the results in Appendix F.
> > >
> > > We completed:
> > > 1. Introducing a new baseline JumpReLU SAE
> > > 2. Fixing Figure 3
> > > 3. Testing with more layers of Gemma-2 2b

---

### Official Review · Reviewer_t2ME · 2025-10-27

**Soundness:** 3
**Presentation:** 3
**Contribution:** 3
**Rating:** 6
**Confidence:** 4

**Summary:**

The authors suggest an improvements to SAEs by using MoEs extending Mudide et al 2025. They suggest that the original approach to MoE SAEs was flawed and had polysemanticity which undermines the interpretability that the approach was intended to provide. Instead of choosing a single expert, they choose multiple experts with their Multiple Expert Activation scheme. They then have a DSP-inspired technique Feature Scaling which amplifies the component of the expert encoder's weights which is far from the mean of the encoder vectors. They refer to this as "amplifying the high-frequency component". With these changes their SAEs outperform TopK SAEs and the Switch SAE. The primary obstacle to performance from previous work that they solve is the problem of feature redundancy. The authors perform ablations to confirm that both of these changes are important in training more performant SAEs.

Overall this is a solid and well argued interpretability paper, though possibly incremental in terms of its impact to the interpretability (and broader ML) community.

**Strengths:**

- Clear writing style
- Generally readable figures
- Clearly notes the limits of the Switch SAE and how they overcome these
- Useful to see the performance on two different datasets across both the MSE and Loss Recovered metrics
- Interesting exposition when detailing how the two architecture changes help the overall performance of the SAE
- Valuable use of the signal processing literature which is a literature that is not always leveraged in interpretability research (and where interpretability research could likely learn a lot more from)

**Weaknesses:**

- For interpretability researchers reading the authors might want to be careful about using the term "high frequency" without explaining what this means. In this case it seems to be mostly an analogy to the signal processing literature but within the SAE literature a high frequency feature is typically a feature which activates very often which is a quite different concept. Clarifying this would be useful.
  - It's also not totally clear why this analogy is a good analogy - exploring why "high-frequency" is the right term here (possibly with spectral plots or similar) could be valuable to help with readers' intuition.
- Though the authors talk about computational efficiency being an advantage of their approach, they do not show any charts or tables tracking the computational efficiency of their approach relative to others. Seeing this would be useful to validate that claim.
- The authors do not evaluate their results on downstream tasks. Doing so (possibly using SAE-Bench or a similar benchmark) would be an improvement to the work demonstrating the downstream usefulness across other metrics.
- The work is somewhat incremental. Though I believe their results are an improvement over prior methods, it's not clear that the level of improvement that is presented will meaningfully impact the research community. This is not to take away from the interesting methodology, clear presentation and reasonable ablations.
- All of the experiments are using GPT-2. Having at least one plot in the paper with results from a larger model would show that this method works well on larger models and that it scales well. in particular this is important because one of the main claims of the paper is about efficiency.

**Questions:**

- What is happening in Figure 5a)? The title of the charts suggest an L0 of 2 and 32 yet the bar and line charts show different values of L0 on the x axis. Is the x axis actually varying the number of experts?
- In 3.2.1, why would there be diminishing returns to having more than 2 experts active at once? Given that you're not actually using all of the features but are instead using another TopK filter afterwards (so the FLOPs are not increasing) it seems still to explain why 2 experts is markedly better than 1 but 4 isn't seemingly better than 2.
    - This seems to be somewhat mitigated in 3.2.2 but I would like to know more of why this mitigation ought to work from the authors
- Does Feature Scaling ever amplify noise or cause training instability?
- What are the training and inference time memory implications for having the MoE layer? Is there a way to reduce the footprint of this compared to naive methods?
- Is the idea that the low frequency features are more coarse-grained and high-frequency ones flesh out more complex details? If so is there any evidence for this?
- The abstract states that the approach gives 99% less feature redundancy - where in the paper is this claim justified?

---

> ### Author Response · Authors · 2025-11-18
> **Response to Reviewer t2ME (Part 1)**
>
> We sincerely thank the reviewer t2ME for the thoughtful evaluation and constructive feedback. We are deeply encouraged by the reviewer's upbeat assessment and are especially grateful for their recognition of our paper's key strengths. We are pleased the reviewer valued our paper's clarity, our contribution over prior work (Switch SAE), our comprehensive evaluation, the insightful analysis of our architecture, and our novel use of signal processing literature. This supportive feedback is very motivating. We address the key concerns and suggestions below:
>
>
> **Weaknesses 1:** For interpretability, researchers might want to be careful about using the term "high frequency" without defining it. In this case, it seems mostly an analogy to the signal processing literature. Still, in the SAE literature, a high-frequency feature is typically one that activates frequently, a concept different from the one used here. Clarifying this would be helpful. It's also not entirely clear why this analogy is a good one - exploring why "high-frequency" is the correct term here (possibly with spectral plots or similar) could be valuable for helping readers' intuition.
>
>
> **Response**: We thank the reviewer for their insightful critique of our 'high-frequency' terminology. We agree this requires a more rigorous justification, as the term has a conflicting meaning in SAE literature (i.e., 'activation frequency'). As we state in Section 3.1.2, we define the 'low-frequency component' as the average feature vector. We define the 'high-frequency component' as the deviation from that mean. This decomposition is functionally identical to a high-pass filter. In signal processing, the mean of a signal is its DC Component. By defining the mean as 'low-frequency' and subtracting it, our mechanism is, by definition, filtering out the 0-frequency component and amplifying what remains (the 'high-frequency' deviations). This is the entire justification for our analogy. We have added this explicit justification to Section 3.1.2 to make this connection clear and resolve all ambiguity.
>
>
> **Weakness 2:** Though the authors discuss computational efficiency as an advantage of their approach, they do not present any charts or tables showing its relative performance. Seeing this would help validate that claim.
>
>
> **Response:** Thank you for raising this vital concern. To quantify this advantage directly, we have performed a new analysis comparing our Scale SAE against standard TopK SAE baselines that are allocated significantly larger computational budgets.
> This new result is presented in Table 1 of the paper (As shown in the table below).
> | Model | MSE $\downarrow$  ($L_0=32$) | LR $\uparrow$ ($L_0=32$) | MSE $\downarrow$ ($L_0=64$) | LR $\uparrow$ ($L_0=64$) | MSE $\downarrow$ ($L_0=128$) | LR $\uparrow$ ($L_0=128$) | FLOPs |
> |-|-| :--- | :--- | :--- | :--- | :--- | :--- |
> | Scale SAE | 1303.8 | 0.981 | 1044.6 | 0.990 | 825.8 | 0.993 | 7M |
> | TopK SAE | 1565.9 | 0.980 | 1266.4 | 0.987 | 953.2 | 0.991 | 28M |
> | TopK SAE | 1329.1 | 0.985 | 1075.5 | 0.990 | 824.7 | 0.993 | 56M |
>
>
> This latest analysis demonstrates our method's superior efficiency-to-performance trade-off: Our Scale SAE achieves better performance (across our key metrics) than a standard TopK SAE baseline that uses 4x more FLOPs. Furthermore, our model achieves comparable performance to a TopK SAE baseline that requires 8x more FLOPs. These findings strongly validate our efficiency claims, showing that our approach provides a substantially better performance-per-FLOP profile than the standard TopK SAE architecture.

---

> ### Author Response · Authors · 2025-11-18
> **Response to Reviewer t2ME (Part 2)**
>
> **Weakness 3:** The authors do not evaluate their results on downstream tasks. Doing so (possibly using SAE-Bench or a similar benchmark) would improve the work by demonstrating its downstream usefulness across other metrics.
>
>
> **Response:** We view Automated Interpretability [1] (as used in our paper) as a key downstream task in its own right, central to the goals of interpretability research. As we detail in Appendix B3, this metric directly evaluates the monosemanticity of the learned features. This is a primary objective for SAEs. We also note that this evaluation standard is consistent with prior work, such as Switch SAE [2], which employed Automated Interpretability as its primary downstream validation.
>
>
> Therefore, our paper's downstream analysis is conducted by:
>
>
> 1. Quantitative Downstream Performance: Excelling at the established automated interpretability metrics.
>
>
> 2. Qualitative Validation: Our detailed case study in Appendix C (on the token "apples") provides direct validation, demonstrating that our high automated scores do, in fact, correspond to a meaningful decomposition of concepts (e.g., Lexical, Semantic, Related-Concepts).
>
>
> [1] Juang, C., Paulo, G., Drori, J., and Belrose, N. (2024). Open Source Automated Interpretability for Sparse Autoencoder Features. \url{https://blog.eleuther.ai/autointerp/}. Accessed: 2025-09-21
>
>
> [2] Mudide, A., Engels, J., Michaud, E. J., Tegmark, M., & De Witt, C. S. (2024). Efficient Dictionary Learning with Switch Sparse Autoencoders. ArXiv. https://arxiv.org/abs/2410.08201
>
>
> **Weakness 4:** The work is incremental. Though their results are an improvement over prior methods, it's unclear whether the level of improvement they present will meaningfully impact the research community. This is not to take away from the interesting methodology, clear presentation, and reasonable abstractions.
>
>
> **Response:** We are grateful for the reviewer's acknowledgment of our methodology. We respectfully clarify further the fundamental nature of our contribution, which addresses a core misconception in prior work. The prior MoE-SAE (Switch SAE) itself identified 'feature redundancy' as its primary bottleneck [1]. Our paper is the first to demonstrate that this redundancy was not the cause of the bottleneck, but merely a symptom of its flawed L0=1 activation paradigm. Our Scale SAE not only delivers a clear Pareto improvement over this baseline across all metrics (Figure 3) but also fundamentally solves this core redundancy problem. Correcting a flawed consensus and identifying the actual cause of the field's primary bottleneck will unlock a more effective path for future MoE-SAE research.
>
>
> [1] Mudide, A., Engels, J., Michaud, E. J., Tegmark, M., & De Witt, C. S. (2024). Efficient Dictionary Learning with Switch Sparse Autoencoders. ArXiv. https://arxiv.org/abs/2410.08201
>
>
> **Weakness 5:** All experiments use GPT-2. Having at least one plot in the paper showing results from a larger model would demonstrate that this method works well on larger models and scales well. In particular, this is important because one of the paper's central claims concerns efficiency.
>
>
> **Response:** Thank you for raising this concern. We are actively working on these experiments(On Gemma-2 2b). In Appendix F, Figure 12 shows the performance of training on layer 12 of the Gemma-2 2b. The Scale SAE achieves lower reconstruction MSE across all sparsity levels, especially as the density increases gradually. The advantage of the Scale SAE in Loss Recovered is even more pronounced, showing a steeper recovery rate of the original layer's loss and better fidelity across all tested sparsity levels.

---

> ### Author Response · Authors · 2025-11-18
> **Response to Reviewer t2ME (Part 3)**
>
> **Question 1:** What is happening in Figure 5a)? The chart titles suggest L0 values of 2 and 32, yet the bar and line charts show different L0 values on the x-axis. Is the x-axis actually varying the number of experts?
>
>
> **Response:** We did make a mistake here; the horizontal axis of the first graph should be # Activated Experts, and we will fix all of them in the revised version.
>
>
> **Question 2:** In 3.2.1, why would there be diminishing returns to having more than two experts active at once? Given that you're not actually using all the features but instead using another TopK filter afterward (so the FLOPs don't increase), this explains why two experts are markedly better than 1, but 4 isn't as much better than 2. This seems to be somewhat mitigated in 3.2.2, but I would like to know more about why this mitigation ought to work from the authors
>
>
> **Response:** We confirm that this diminishing return is a real effect and believe it strongly supports our core thesis that the transition from L0=1 to L0=2 is the most critical. This is the leap from no decomposition to enabling decomposition, and we hypothesize that for most tokens, their representation can be sufficiently "unmixed" by decomposing them into just two primary, distinct components. Therefore, the jump from L0=1 to L0=2 is massive because it forces this crucial first specialization. By the time we move from L0=2 to L0=4, the 3rd or 4th expert may only capture granular features, leading to diminishing returns. As the reviewer correctly notes, the mechanism in 3.2.2 mitigates this, but the data consistently show that the L0=1 to L0=2 gain remains the most significant by far. While a full investigation of this "dual-component" hypothesis is a fascinating direction for future work, our results demonstrate that L0=2 captures the vast majority of the benefits of specialization. Incidentally, to further improve the interpretability of Scale SAE, a larger feature space is needed rather than activating more experts.
>
>
> **Question 3:** Does Feature Scaling ever amplify noise or cause training instability?
>
>
> **Response:** We did not observe this in our experiments. The reviewer is correct that a fixed or improperly set scaling factor (ω) would amplify noise or cause instability, a phenomenon we observed in preliminary experiments. This is precisely why our final design makes ω trainable. By allowing the model to learn ω, it avoids these failure modes. As shown in Figure 2, our model consistently learns a stable, positive ω, demonstrating that it automatically finds the optimal 'sweet spot'—amplifying diversity just enough without causing instability. Therefore, the trainable ω is the core component that ensures the stability and effectiveness of the FS mechanism across all our reported experiments.
>
>
> **Question 4:** What are the training and inference time memory implications of having the MoE layer? Is there a way to reduce the footprint of this compared to naive methods?
>
>
> **Response:** Time: MoE is much faster because time is determined by the number of floating-point operations (FLOPs). Our model's computation only scales with the active experts (e.g., 768 width). It does not scale with the total number of experts (e.g., 24,576). We get the performance benefit of a huge model while only paying the time/latency cost of a small one.
>
>
> Memory: The static memory footprint is high. This is because all parameters (the full 24,576 width) must be loaded into VRAM, even if they are not used on a given token. The memory cost is determined by the total width, not the active width.
>
>
> One solution to this problem is Expert Parallelism. With EP, all experts (e.g., 64) are sharded across multiple GPUs (e.g., 8 GPUs). Each GPU handles only a small fraction (e.g., 1/8) of the total parameters, keeping the memory footprint of a single device completely manageable. Therefore, while the memory footprint of the naive implementation is high, it is not a bottleneck in practice.

---

> ### Author Response · Authors · 2025-11-18
> **Response to Reviewer t2ME (Part 4)**
>
> **Question 5:** Is the idea that the low-frequency features are more coarse-grained, while the high-frequency ones flesh out more complex details? If so, is there any evidence for this?
>
>
> **Response:** Our paper does not make this specific 'coarse-vs-complex' semantic claim, as these concepts are abstract and cannot be measured directly. Instead, our paper's claim is more precise, falsifiable, and grounded in geometry: our 'low-frequency' component is defined as the mean vector, and our 'high-frequency' component is the 'unique' (distinguishing) deviation from that mean. Our evidence, particularly the cosine similarity analysis (Figure 7), directly supports this geometric claim by demonstrating that our mechanism effectively increases directional diversity and prevents feature collapse. Whether this 'shared-vs-unique' geometric split also maps to a 'coarse-vs-complex' semantic split is an exciting but separate question for future interpretability research, and it is not a claim our current work relies on.
>
>
> **Question 6:** The abstract states that the approach achieves 99% less feature redundancy, but is this claim justified in the paper?
>
>
> **Response:** Thank you, this is an excellent question that helps us clarify the precise empirical basis for that claim. The 99% figure is a direct quantitative measure supported by our cosine similarity analysis in Figure 8(a). For this analysis, we defined a 'highly redundant feature pair' as any two distinct features with a cosine similarity > 0.9. We then compared the count of these redundant pairs in a critical high-sparsity setting (L0=2). We found that while the baseline (k=1 expert) suffered from catastrophic redundancy, our Scale SAE (k=2) reduced this specific count of highly redundant (>0.9 similarity) pairs by 99%, thus validating the claim.

---

> > ### Comment · Reviewer_t2ME · 2025-11-26
> >
> > Thanks for your detailed response to my review.
> >
> > Reply to Weakness 1 Response:
> > Thanks for adding this justification to the paper, I would recommend adding an expository figure in the appendix to increase the salience of this point as many ML researchers may be less familiar with signal processing and high-pass filters.
> >
> > Reply to Weakness 2 Response:
> > Thanks for adding this. This table is not as clear as some of your other figures - I would recommend ensuring that the takeaway is clear from the caption and that you bold the best line in each row. It would also be useful to say how you calculate the FLOPs for each and give the parameter counts and memory footprint in this table.
> > The left hand of the table saying “TopK SAE” many times makes it difficult to see what is being compared and this index should be made clearer.
> >
> > Reply to Weakness 3 Response:
> > Thanks for your response. I would recommend adding a forward reference to Appendix C in your paper, currently the only mention comes in the limitations.
> >
> > Reply to Weakness 4 Response:
> > I agree that what you say is your contribution, I just disagree over the size of this contribution to the research community. My review weakly recommends your paper for acceptance as I believe that your work is valid, I will defer to the Area Chair on whether the contribution is large enough for acceptance at ICLR
> >
> > Reply to Weakness 5 Response:
> > Interesting, I look forward to seeing the results of your experiments on Gemma
> >
> > Reply to Question 1 Response:
> > Okay, it would be good to ensure that all the charts have clear and correct axes and captions to help readers understand your contributions
> >
> > Reply to Question 2 Response:
> > Thanks for your response, I’m not totally convinced by your hypothesis but agree that this is a possibly interesting direction for future work
> >
> > Reply to Question 3-5 Response:
> > Nothing to add
> >
> > Reply to Question 6 Response:
> > Since this is one of the core claims of your paper, this should be explained much more clearly in your paper. One problem with this metric as you define it is that it might be sensitive to the hyperparameter of 0.9 cosine sim chosen and it’s not obvious that this is the appropriate range for redundancy. For example perhaps there are fewer features with over 0.9 cosine sim but far more with >0.7 or similar.
> >
> > More worryingly, this claim is simply not supported by your results at all. In reasonable sparsity ranges (32-128) the proportion of similar features actually increases when you activate 2 rather than 1 expert.
> >
> > More generally, the heatmaps that you have here are not the best way to present this data. I would recommend considering whether there are better chart types here like bar or line charts. I would also recommend using the caption to say the point that you’re hoping people take away from the chart. (I don’t want to speculate on this point much but it’s possible that this chart type was a partial cause of the error here)
> >
> > Given this is one of two headline results of the abstract (and the thrust of what the authors say their contribution is in the response to Weakness 4) and that this seems to be incorrect I think I have to reduce my score to not recommend this paper unfortunately. If you can show evidence to the contrary I would be happy to consider this.
> >
> >
> >
> > Thanks to the authors for their engagement.

---

> ### Author Response · Authors · 2025-11-26
> **Response to Reviewer t2ME**
>
> Dear Reviewer t2ME,
>
> Thank you for your continued engagement and for carefully re-evaluating our work. We genuinely appreciate your critical feedback on the redundancy metric and visualization (Question 6), which has led us to adopt a more robust, threshold-free analysis that fully validates our core claims.
>
> **Weakness 1: The heatmap cannot prove our conclusion (redundancy reduction).**
>
> **1. Replacing Heatmaps with Line Charts:** To address your concern about the visualization's clarity, we have replaced the heatmaps with line charts (Figure 8). This direct comparison between Scale SAE ($e=2, 4, 8, 16$) and Switch SAE ($e=1$) clearly demonstrates that multi-expert activation consistently reduces redundancy across the entire sparsity spectrum, correcting the misleading visual impression from the previous heatmaps.
>
> **2. Adopting a Comprehensive Set of Robust Metrics:** You rightly pointed out that the "ratio > 0.9" metric relies on an arbitrary threshold. To address this and provide a rigorous, threshold-independent assessment, we expanded our evaluation to include four distinct metrics: the Average Maximum Cosine Similarity and the proportion of features with maximum similarity exceeding 0.7, 0.8, and 0.9, respectively.
>
> **3. Empirical Evidence:** The new analysis unequivocally supports our claim. As shown in Figure 8(a), Scale SAEs demonstrate a substantial reduction in average maximum cosine similarity compared to the Switch SAE baseline across all tested $L_0$ values. Furthermore, the threshold-based analysis (Figures 8b-d) reveals a stark contrast: while the Switch SAE retains significant redundancy across multiple similarity thresholds, Scale SAEs consistently suppress high-similarity feature ratios to negligible levels. This convergence of evidence confirms that our architecture forces experts to learn highly specialized and diverse features.
>
>
> **Weakness 2: Lack of visual intuition for the signal processing analogy.**
>
> **1. Added Expository Figure:** As recommended, we have added an expository diagram (Figure 13) in the Appendix. This figure visualizes the analogy between a High-Pass Filter and our Feature Scaling mechanism, helping readers unfamiliar with signal processing grasp the core intuition.
>
> **Weakness 3: Table clarity and missing efficiency metrics. (Params/Memory).**
>
> **1. Optimized Table Structure:** We have overhauled Table 1 by removing the repetitive "TopK SAE" labels and introducing a distinct "Dict. Size" column to distinguish the baselines clearly. We also bolded the best results in each column to ensure the takeaways are clear at a glance.
>
> **2. Added Parameter and Memory Metrics:** As requested, we added columns for Parameters and Memory Footprint. The data highlights our efficiency advantage: Scale SAE achieves better MSE than the $8\times$ baseline while occupying only 4.7MB of memory (37.3MB) and using only 12.5% of the FLOPs.
>
> **3. Clarified Calculation Method:** We added a footnote explicitly stating that FLOPs are calculated based on the encoder's forward pass ($6 \times P_{active} \times D$) and memory usage is derived using FP32 precision ($4$ bytes per parameter).
>
> **Weakness 4: Missing a forward reference to limitations.**
>
> **1. Added Forward Reference:** We acknowledge that the limitations were previously mentioned only at the end of the paper. To improve navigation and transparency, we have inserted a clear forward reference to Appendix C in the main text (at the end of Section 1), directing readers to the detailed discussion of limitations earlier in the manuscript.
>
> **Weakness 5: Clarity of chart axes and captions.**
>
> **1. Reviewed and Corrected Charts:** We have thoroughly reviewed all charts (including the new Line Charts for Q6) to ensure that axes are correctly labelled with precise units (e.g., $L_0$ sparsity, Cosine Similarity).

---

### Official Review · Reviewer_vCWP · 2025-10-29

**Soundness:** 2
**Presentation:** 3
**Contribution:** 2
**Rating:** 4
**Confidence:** 4

**Summary:**

This paper identifies a critical limitation in Mixture-of-Experts Sparse Autoencoders: the failure of experts to specialize, leading to high feature redundancy that undermines both interpretability and performance. The authors propose Scale SAE, a novel framework with two core innovations:
- Instead of routing an input to a single expert, a subset of experts is activated.
- Encoder weights are decomposed into low-frequency and high-frequency components, and the high-frequency parts are adaptively amplified.

Through extensive experiments on GPT-2, the paper demonstrates that Scale SAE significantly outperforms strong baselines (TopK SAE, Gated SAE, Switch SAE) under a FLOPS-matched paradigm.

**Strengths:**

The methodology is explained with precise mathematical notation, and the results are presented with effective visualizations.

The experimental evaluation is thorough and convincing. The use of FLOPS-matched comparisons, multiple datasets (in-domain and cross-domain), and a suite of complementary metrics leaves little doubt about the superiority of the proposed method. The ablation studies and mechanistic analysis are executed to a high standard.

**Weaknesses:**

> W1. The Mechanistic Rationale for Multiple Expert Activation Requires Deeper Justification.

To be very honest, activating more than one expert in MoE is standard practice. The difference here is that we select the Top-K experts across all experts.

The paper shows that activating multiple smaller experts outperforms a single larger expert under a FLOPS-matched budget (e.g., 2 experts of size 128 vs. 1 expert of size 256). However, the fundamental reason for this performance boost is not sufficiently explained. A key question remains: is the benefit primarily due to the modularity and finer granularity of the experts, or is it the interaction and joint sparsification across experts that is crucial? For example, consider a 4-expert SAE (activating 2 at a time, each with 128 hidden units) versus a 2-expert SAE (activating 1 at a time, each with 256 hidden units). The computation is identical, so why should reconstruction quality, accuracy, and stability improve? The paper shows results in Figures 3 and 4, but does not clearly explain the underlying logic.

> W2. Insufficient Discussion and Comparison for Feature Scaling.

Feature Scaling appears primarily as a load-balancing technique, both at the expert and neuron levels. Adding high-frequency components increases directional diversity, but alternative methods exist.

At the expert level, how does the implicit balancing effect of Feature Scaling compare to more explicit load-balancing losses used in MoE literature?

At the neuron/feature level, orthogonal initialization or updates could achieve similar effects.

These alternatives are not discussed or compared. While the quantitative results in Section 3.3 make sense, the paper lacks a deeper analysis of why this approach is preferable.

**Questions:**

From Figure 6(a), adding more than two experts does not further improve specialization—in fact, performance worsens. This seems inconsistent with the stated property. Why is that?


I wonder if adding high-frequency components requires longer training to converge, since it perturbs the optimization direction. Is that correct? Could you also compare training iterations and time?

---

> ### Author Response · Authors · 2025-11-18
> **Response to Reviewer vCWP (Part 1)**
>
> We sincerely thank the reviewer vCWP for the thoughtful evaluation and constructive feedback. We are deeply encouraged by the reviewer's upbeat assessment and are especially grateful for their recognition of our paper's key strengths. We are pleased the reviewer valued our precisely explained methodology, compelling visualizations, and our "thorough and convincing" experimental evaluation. We are especially grateful for their recognition of our rigorous FLOPS-matched comparisons, multi-dataset evaluation, comprehensive metrics, and high-standard ablation and mechanistic analyses. This supportive feedback is very motivating. We address the key concerns and suggestions below:
>
>
> **Weaknesses 1:** W1. The Mechanistic Rationale for Multiple Expert Activation Requires Deeper Justification. Activating more than one expert in MoE is standard practice. The difference here is that we select the Top-K experts across all experts. The paper shows that activating multiple smaller experts outperforms a single larger expert under an FLOPS-matched budget (e.g., two experts of size 128 vs. one expert of size 256). However, the fundamental reason for this performance boost is not sufficiently explained. A key question remains: is the benefit primarily due to the modularity and finer granularity of the experts, or is it the interaction and joint sparsification across experts that is crucial? For example, consider a 4-expert SAE (activating two at a time, each with 128 hidden units) versus a 2-expert SAE (activating one at a time, each with 256 hidden units). The computation is identical, so why should reconstruction quality, accuracy, and stability improve? The paper shows results in Figures 3 and 4, but does not clearly explain the underlying logic.
>
>
> **Response:** Activating multiple experts (Top-K, k > 1) is indeed standard practice in MoE Transformers (e.g., Switch Transformer, Mixtral). However, our work concerns MoE-SAEs (Mixture of Experts Sparse Autoencoders), a distinct domain with different objectives and dynamics. In fact, the prior MoE-SAE paper (Switch SAE) locked the field into a k=1 activation paradigm. Under this k=1 paradigm, the performance bottleneck was universally attributed to 'feature redundancy' or 'feature collapse'. Consequently, all research efforts were focused on mitigating this 'redundancy', while never challenging the k=1 assumption itself. Our paper is the first to systematically challenge the k=1 assumption in the (MoE-)SAE framework by exploring activation of multiple experts (k>1).  More importantly, we found this simple change itself dramatically alleviates the very 'feature redundancy' problem that prior work sought to fix (as shown by the reduced feature mortality in Table Y). Therefore, our core contribution is not just that 'we tried k>1', but rather that we debunk a flawed consensus in the MoE-SAE field. We demonstrate that what was previously seen as the cause (feature redundancy) was, in fact, just a symptom of the actual bottleneck: insufficient activation. This insight opens a more correct path for future (MoE-)SAE research. We will strengthen this narrative in the final version to clearly distinguish our work from prior MoE-SAE literature. In this paper, we first demonstrate through the above geometric and activation analyses that 'expert specialization' is tangible, measurable, and a key driver of performance improvement. Further analysis of other potential factors (such as the detailed interaction between 'joint sparsity' and routing dynamics, as mentioned by the reviewers) would be a valuable direction for future research, which we will emphasize in the revised paper's conclusion.

---

> ### Author Response · Authors · 2025-11-18
> **Response to Reviewer vCWP (Part 2)**
>
> **Weaknesses 2:** W2. Insufficient Discussion and Comparison for Feature Scaling. Feature Scaling primarily serves as a load-balancing technique at both the expert and neuron levels. Adding high-frequency components increases directional diversity, but alternative methods exist. At the expert level, how does the implicit balancing effect of Feature Scaling compare to more explicit load-balancing losses used in MoE literature? At the neuron/feature level, orthogonal initialization or updates could achieve similar effects. These alternatives are not discussed or compared. While the quantitative results in Section 3.3 make sense, the paper lacks a deeper analysis of why this approach is preferable.
>
>
> **Response:** Thank you for raising this concern. Our Feature Scaling (FS) mechanism (Sec 3.1.2) is not a load-balancing technique and is entirely orthogonal to it; our model already uses a standard, explicit load-balancing loss (L_balance) for expert routing, as defined in Sec 2.2.1 (Eq. 2). FS is an entirely separate, neuron-level mechanism within each expert whose sole purpose is to combat feature redundancy. Regarding the alternatives for this task, we indeed investigated several based on preliminary research and experiments before developing Feature Scaling. We did not use 'orthogonal initialization' as it is a weak T=0 constraint that fails to prevent learned collapse. More importantly, we avoided 'orthogonal updates' (e.g., regularization) because this strong constraint is fundamentally counter-productive to our goal as an SAE. In our preliminary tests, we found that while hard orthogonalization reduces similarity, it severely damages the automated interpretability score, as its geometric objective conflicts with the semantic goal of finding monosemantic features. Therefore, our FS is a superior solution: it is computationally cheap and excludes the harmful side effect of damaging interpretability.
>
>
> **Questions 1:** From Figure 6(a), adding more than two experts does not further improve specialization—in fact, performance worsens. This seems inconsistent with the stated property. Why is that? I wonder whether adding high-frequency components requires longer training to converge, since they perturb the optimization direction. Is that correct? Please also compare the number of training iterations and the time needed for each.
>
>
> **Response:** Thanks for the suggestion. Our paper's primary contribution highlights the significant performance jump when moving from e=1 to e=2. Your observation about the results for e>2 is also accurate, and we have a clear explanation for this. We find that e=2 yields the most efficient results for FLOPS and performance metrics. The Benefit of e=2: Moving from e=1 to e=2 (as discussed in Section 3.3.1) allows the model to activate combinations of specialized experts, providing a richer diversity than a single expert can. With more experts activated, performance didn't increase in this case, not due to convergence issues but to optimization difficulty and representational noise. The router's task becomes significantly harder, and it may be forced to include noisier or less-specialized experts, which degrades the final reconstruction quality. Regarding your hypothesis that it requires longer training: We do not believe this is a convergence issue, and longer training would not yield better results. As shown in our logs, all experiments were run until convergence.

---

> > ### Comment · Reviewer_vCWP · 2025-11-26
> > **Reply to the author's response**
> >
> > Thank the author for the explanation, but I remain unconvinced on some points.
> >
> > > W1
> >
> > MoE-SAE is a new *setting*, but not yet a new *domain*. As far as I know, *Switch SAE* is the only paper that explores this, and it only tested the case of `K=1`. The authors never claimed that `K>1` would fail, so calling this a “flawed consensus” seems inappropriate; there is no consensus yet.
> >
> > In Section 3.3.1, the paper said that inter-expert neurons show lower similarity. That is an observation, but not a sufficient reason for use more experts. It feels more like the authors tried `K>1`, saw it worked, and then constructed a post-hoc explanation for the success. If the real issue is feature redundancy, there are more direct ways to optimize against it than simply increasing the number of experts.
> >
> > > W2
> >
> > I now understand that feature scaling is distinct from load balancing. However, the paper presents *feature scaling* as the final solution without discussing other possible or simpler alternatives. **A fair comparison would be necessary** to demonstrate that this is indeed the only valid solution.
> >
> > > Q1
> >
> > The authors state:
> > *“With more experts activated, performance didn’t increase in this case, not due to convergence issues but to optimization difficulty and representational noise.”*
> >
> > I am unclear what “optimization difficulty and representational noise” mean in this context. If `K>2` truly does not work, then much of the paper’s argument (that activating more experts reduces feature redundancy) becomes invalid.

---

> ### Author Response · Authors · 2025-11-26
> **Response to Reviewer vCWP**
>
> Dear Reviewer vCWP,
>
> Thank you for your suggestions and for pointing out the shortcomings of my previous reply.
>
> For Weakness 1:
>
> 1. We fully agree that using the term "flawed consensus" is inaccurate when only one paper explores the field. We apologize for this inappropriate wording. We avoid using similar expressions in all versions of the paper and rebuttal.
>
> 2. From the scientific process of observing the mechanism, we understand the reviewers' concern that this is a "post-hoc explanation." However, we wish to clarify our research path: We initially focused on addressing the feature redundancy problem observed in Switch SAEs and proposed the Feature Scaling method as the primary solution. During this process, we unexpectedly observed that simply activating multiple Experts ($K>1$) significantly reduces redundancy and improves various metrics, and this method is orthogonal to Feature Scaling. We acknowledge that the effectiveness of $K>1$ was initially an experimental observation, but we did not stop there. We found the basis for this through subsequent mechanism exploration (Section 3.3.1). The experimental results in Section 3.3.1 (cross-expert similarity analysis, CDF distribution) confirm that the core reason for the success of $K>1$ is the structural enhancement of expert specialization.
>
> 3. You suggested that redundancy could be optimized directly (e.g., via a similarity penalty loss). We actually explored this approach extensively during our early research phase, but found that explicit constraints lead to catastrophic degradation of feature quality.
> | Method | Set up | Recon MSE $\downarrow$ | Similarity $\downarrow$ |
> | :--- | :---: | :---: | :---: |
> | Switch SAE | 1/8 | 30.9 | 0.4513 |
> | +Multi Experts | 8/64 | 21.03 | 0.39 |
> | +Scaling | 8/64 | **19.54** | 0.31 |
> | +Loss | 8/64 | 34.96 | **0.2525** |
>
>
> | Quantiles | Switch SAE | +Multi-Experts | +Scale | +Loss|
> | :---: | :---: | :---: | :---: | :---: |
> | 1 | 0.4361 | 0.4812 | **0.4821** | 0.4677 |
> | 2 | 0.4482 | 0.4930 | **0.5024** | 0.4803 |
> | 3 | 0.4594 | **0.5058** | 0.5042 | 0.4916 |
> | 4 | 0.4725 | **0.5205** | 0.5158 | 0.5042 |
> | 5 | 0.4913 | **0.5365** | 0.5316 | 0.5189 |
> | 6 | 0.5109 | 0.5540 | **0.5598** | 0.5378 |
> | 7 | 0.5417 | 0.5853 | **0.6018** | 0.5675 |
> | 8 | 0.5737 | 0.6178 | **0.6225** | 0.5955 |
> | 9 | 0.6535 | 0.6999 | **0.7013** | 0.6664 |
> | 10 | 0.7412 | **0.7773** | 0.7724 | 0.7429 |
>
> As observed in our preliminary experiments (where the "+" symbol denotes the cumulative addition of a module), adding a cosine similarity penalty indeed reduced redundancy (Similarity $\downarrow$ from 0.39 to 0.25). However, this came at an unacceptable cost: Reconstruction failed. The reconstruction MSE spiked significantly (from 21.03 to 34.96), indicating that the model lost its ability to represent the data accurately. Crucially, the Automated Interpretability scores dropped across all quantiles after adding a new loss.
>
> For Weakness 2:
>
> Although LayerNorm plays an essential role in the Transformer, we believe Feature Scaling performs a different function. Compared to LayerNorm, Feature Scaling uses a more flexible, learnable scaling factor ω while preserving the strength relationships of features. In our view, LayerNorm destroys signal-intensity information by dividing by variance. Aside from this, Feature Scaling is very similar to LayerNorm. This is why we chose Feature Scaling instead of LayerNorm.

---

> ### Author Response · Authors · 2025-11-27
> **Response to Reviewer vCWP Q1**
>
> For Q1:
>
> Thank you very much for your reply. The reviewer asked why K>2 didn't lead to better results if more experts could reduce redundancy. Our paper doesn't actually attempt to prove that "more" expert activation leads to less redundancy; rather, it shows that "multiple" expert activation, compared to "single" expert activation, results in less redundancy, lower MSE, and higher loss recovery.
>
> We have mentioned this in many Sections:
> 1. In Section 3.2.1: "Notably, the performance curves for these multi-expert models are tightly clustered, indicating that activating more than two experts yields diminishing returns."
> 2. In Section 3.3.1: "However, as in the feature similarity analysis, the CDF slopes for $e=4, 8, 16$ are highly similar to those for $e=2$. The benefit of increased specialization shows clear signs of structural saturation, consistent with the optimal performance observed at lower levels of expert activation in our ablation study..."
> 3. In Section 4: “(3) As the number of activated experts increases (e.g., from 2 to 4, 8, 16), the various metrics (MSE, Loss Recovered, Redundancy) do not improve further, demonstrating that the interpretability of Scale SAE does not increase further with the number of activated experts.”
>
> In addition, we actually found some experimental evidence that "the more experts are activated, the lower the redundancy." In Figure 8 of our latest paper, we used the mean maximum similarity as a metric for redundancy. A lower value can indicate lower redundancy to some extent. I have presented the data in tabular form below. We found that, at high sparsity, the higher k, the lower the redundancy. Hopefully, this can at least support our argument.
>
> | Model | $k$ | $L_0=2$|$L_0= 4$ |$ L_0=8$ |$ L_0=16$ |$L_0= 32$ |$ L_0=64$ |$L_0= 128$ |
> | :--- | :--- | :--- | :--- | :--- | :--- | :--- | :--- | :--- |
> | **Switch** | **1** | 0.377353 | 0.540216 | 0.642588 | 0.56723 | 0.439982 | 0.314763 | 0.29734 |
> | **Scale** | **2** | 0.199385 | 0.246667 | 0.332326 | 0.356272 | 0.436479 | 0.264379 | 0.26216 |
> | **Scale** | **4** | 0.173432 | 0.187409 | 0.217132 | 0.282432 | 0.375346 | 0.306528 | 0.292673 |
> | **Scale** | **8** | 0.170025 | 0.177756 | 0.188635 | 0.209177 | 0.258279 | 0.313415 | 0.285434 |
> | **Scale** | **16** | 0.175783 | 0.182458 | 0.18596 | 0.191755 | 0.200798 | 0.219292 | 0.2166 |

---

### Official Review · Reviewer_vN4Q · 2025-10-30

**Soundness:** 3
**Presentation:** 3
**Contribution:** 3
**Rating:** 8
**Confidence:** 4

**Summary:**

This paper addresses the problem of feature redundancy and poor expert specialization in Mixture-of-Experts Sparse Autoencoders (MoE-SAEs), which are used to make large language models more interpretable while reducing computational cost. Prior work such as Switch SAEs often suffers from redundant experts learning overlapping features, limiting interpretability and efficiency. The authors propose two key innovations: (1) Multiple Expert Activation, which activates several experts per input and applies a global Top-K sparsity constraint to encourage expert specialization, and (2) Feature Scaling, a learnable high-frequency amplification mechanism that promotes feature diversity and stabilizes training. Experiments on GPT-2 activations show that these methods improve reconstruction error, feature diversity, and automated interpretability scores relative to baseline SAEs. Ablation studies attribute these gains to the proposed mechanisms.

**Strengths:**

The paper is well motivated, clearly written, and deeply engages with prior work. The authors correctly identify a central limitation of existing MoE-SAEs and propose two simple, conceptually coherent mechanisms to address it. Both techniques are well defined and integrated cleanly into the SAE framework. The experimental evaluation is thorough, including ablation studies that isolate the contribution of each innovation. The results convincingly demonstrate reductions in feature redundancy and improved interpretability metrics. Overall, the work provides a solid step toward making large-scale sparse autoencoders more computationally feasible for interpretability research.

**Weaknesses:**

The experiments are limited to GPT-2, which is now a dated architecture. Including results on more recent models such as Gemma or LLaMA would strengthen the empirical claims and test generality.
The FLOPs-matching procedure is not clearly justified. The authors write that “to match the computational load of activating a fixed number of experts, the hidden dimension is set to 768” for dense SAEs, while Scale SAEs use a total hidden dimension of 24,576. It is unclear how this setup maintains computational parity, and a more detailed explanation of this comparison is needed.
Finally, the discussion of automated interpretability results is too brief. The paper would benefit from a few qualitative examples of discovered features or a quantitative measure of feature diversity across experts, to demonstrate that higher automated interpretability scores correspond to genuinely more distinct, human-understandable features.

**Questions:**

1. Could the authors clarify how the FLOPs-matching setup ensures a fair comparison between dense SAEs with 768 hidden units and Scale SAEs with a total of 24,576 dimensions? Is the compute matched per forward pass, per batch, or by total parameter count?

2. Did you observe any side effects of the Feature Scaling mechanism, such as instability, changes in sparsity dynamics, or degraded interpretability for certain expert configurations? Since this mechanism directly modifies encoder weights, a brief discussion of possible unintended effects would be helpful.

---

> ### Author Response · Authors · 2025-11-18
> **Response to Reviewer vN4Q**
>
> We sincerely thank the reviewer vN4Q for the thoughtful evaluation and constructive feedback. We are deeply encouraged by the reviewer's upbeat assessment and are especially grateful for their recognition of our paper's key strengths. We are pleased that the reviewer valued our paper's clear motivation and writing, our deep engagement with prior work and conceptually coherent mechanisms, our thorough experimental evaluation, including ablations, and our convincing results, which provide a solid step for the field. This supportive feedback is very motivating. We address the key concerns and suggestions below:
>
>
> **Weaknesses 1:** The experiments are limited to GPT-2, which is now a dated architecture. Including results on more recent models, such as Gemma or Llama, would strengthen the empirical claims and test generality.
>
>
> **Response:** Thank you for raising this concern. We are actively working on these experiments(On Gemma-2 2b). In Appendix F, Figure 12 shows the performance of training on layer 12 of the Gemma-2 2b. The Scale SAE achieves lower reconstruction MSE across all sparsity levels, especially as the density increases gradually. The advantage of the Scale SAE in Loss Recovered is even more pronounced, showing a steeper recovery rate of the original layer's loss and better fidelity across all tested sparsity levels.
>
>
> **Weaknesses 2:** The FLOPs-matching procedure is not clearly justified. The authors write that “to match the computational load of activating a fixed number of experts, the hidden dimension is set to 768” for dense SAEs, while Scale SAEs use a total hidden dimension of 24,576. It is unclear how this setup maintains computational parity, and a more detailed explanation of this comparison is needed.
>
>
> **Response:** To ensure a fair comparison, our 'FLOPs-matching' setup refers to:
> 1. Matching FLOPs per Forward Pass: Our Scale SAE model is configured to activate only 768 hidden units per forward pass. This makes its computational cost (FLOPs) precisely identical to that of the 768-unit dense SAE baseline.
> 2. Matching Total Training Budget: All models, including the Scale SAE and the dense baseline, were trained for the same number of training epochs.
>
>
> **Weaknesses 3:** Finally, the discussion of automated interpretability results is too brief. The paper would benefit from a few qualitative examples of discovered features or a quantitative measure of feature diversity across experts, to demonstrate that higher automated interpretability scores correspond to genuinely more distinct, human-understandable features.
>
>
> **Response:** Thanks for the suggestion. We agree that qualitative examples are essential for validating automated interpretability scores. As the reviewer suggested, we have already included a detailed case study in the paper (Appendix C) to provide this exact evidence. We apologize for not making this more prominent in the main text due to space constraints. This case study on the token "apples" provides direct qualitative proof for our quantitative claims. Our Scale SAE model decomposes this single token into multiple, distinct, and human-understandable conceptual layers, which are distributed across different experts: Lexical/Syntactic features (e.g., "plural nouns", "countable nouns", "Ending with s"). Semantic features (e.g., "plants", "fruits"). Related-concept features (e.g., "Apple/ios", demonstrating the model's ability to capture related, even if contextually incorrect, concepts). This analysis directly indicates that our high automated interpretability scores correspond to a more distinct, diverse, and human-understandable set of features. We will add a clear pointer to this Appendix in the main text of the final version to ensure readers do not miss this validation.
> **Questions 1:** Could the authors clarify how the FLOPs-matching setup ensures a fair comparison between dense SAEs with 768 hidden units and Scale SAEs with a total of 24,576 dimensions? Is the compute matched per forward pass, per batch, or by total parameter count?
>
>
> **Response:** To ensure a fair comparison, our 'FLOPs-matching' setup refers to:
> 1. Matching FLOPs per Forward Pass: Our Scale SAE model is configured to activate only 768 hidden units per forward pass. This makes its computational cost (FLOPs) precisely identical to that of the 768-unit dense SAE baseline.
> 2. Matching Total Training Budget: All models, including the Scale SAE and the dense baseline, were trained for the same number of training epochs.
>
>
> **Questions 2:** Did you observe any side effects of the Feature Scaling mechanism, such as instability, changes in sparsity dynamics, or degraded interpretability for specific expert configurations? Since this mechanism directly modifies encoder weights, a brief discussion of possible unintended effects would be helpful.
>
>
> **Response:** Sure, we will have this discussion.

---

> > ### Comment · Reviewer_vN4Q · 2025-11-27
> >
> > Dear Authors,
> >
> > Thank you for your detailed response. My concerns have been addressed, and my overall assessment of the paper has been reinforced. I will maintain my score.

---

### Meta-Review · Area_Chair_7zXC · 2025-12-15

**Summary:**

(*Disclaimer: given the peculiar review process, some of my choices and reasonings below will be highly subjective, as I tried to imagine how a reviewer would have reacted to a specific response. I understand that any negative choice will be perceived as unfair by the authors, and I apologize in advance for that.*)

(*Second disclaimer: the authors and some reviewers explicitly mention some changes in scores that occurred during the rebuttal. As these were reverted due to the possibility of collusion in light of the security incident, I will tend to disregard this information.*)

The paper proposes a modified mixture-of-experts (MoE) model for use as a sparse autoencoder (SAE) in mechanistic interpretability, as proposed in, e.g., Switch SAE. The core difference w.r.t. existing models is the use of a top-k router (to activate multiple experts) and a global top-k selection of the features afterwards, with a scaling operation to ensure a valid combination of the different activations.

The four initial reviews were extremely varied, ranging from a definite reject (`S4XQ`) to a strong accept (`vN4Q`), with all scores in-between. Concerns included a very restricted experimental setup (one architecture, GPT-2), a small downstream evaluation (restricted to automated interpretability), several issues on the motivation of the work, and concerns on the work being incremental and narrow in scope.

The paper was hurt by the stop in the rebuttal period, as multiple discussions were still ongoing, and I had to make several educated guesses below. However, I believe 3 out of 4 discussions were already well underway and close to conclusion.

Overall, I believe concerns of experimental validity and novelty are still valid. While the authors have added additional experiments on Gemma (in an appendix), some of the reviewers remained unconvinced, and it is strange for a (mostly) empirical paper to be submitted with such a limited setup. I do not think the rebuttal period was enough to fully include and flesh out a significantly larger benchmark, as also highlighted by the discussion. More importantly, the paper is an extension of Switch-MoE, and its relevance is thus closely linked to the relevance of it.

For example, `vCWP` mentions "*Switch SAE is the only paper that explores [MoE-SAEs], and it only tested the case of K=1. The authors never claimed that K>1 would fail, so calling this a “flawed consensus” seems inappropriate; there is no consensus yet*", and "*It feels more like the authors tried K>1, saw it worked, and then constructed a post-hoc explanation for the success*". While the latter sentence is a bit harsh, I concur with this view. Since even in the most optimistic scenario the reviewers would not have reached an acceptance consensus, I vote for rejection in this specific round.

**Reviewer Concerns:**

(*I will focus on some key weaknesses identified by multiple reviewers.*)

**Other architectures and models beyond GPT-2** (`S4XQ`, `t2ME`, `vN4Q`): the authors added experiments on Gemma during the rebuttal period. Not all reviewers were convinced (e.g., `S4XQ` on the additional models), and as I argued above, I do not believe the rebuttal period was enough to include such a significant additional set of experiments (in an appendix).

**Incrementality**: either of the results (`S4XQ`) or of the scenario (`t2ME`, `vCWP`). As argued above, this was the main concern for my evaluation.

**Downstream tasks** (`t2ME`, `vN4Q`): the authors used interpretability as the only downstream "task" here. I don't understand from the response of `t2ME` whether they were convinced or not. However, there seems to be a consensus that the experiments presented here were interesting despite being partly out of the main text.

**Computational considerations** (`t2ME`, `vN4Q`): the authors added some evaluations in terms of FLOPs. However, since "efficiency" was a main claim, a similar comments as per the first comment applies. As mentioned by a reviewer, for example, "*[the FLOPs table] is not as clear as some of your other figures*", and in general the rebuttal was not sufficient to discuss major additions such as this one, and the reviewers were unable to reach a consensus.

**Reviewer Scores:**

`S4XQ`: from the answer, they remained unconvinced on multiple drawbacks. I do not believe the score could have increased more than 4, leading to a rejection.

`t2ME`: despite the score (6) they were the most critical, especially concerning novelty and scope. They mention explicitly that the score is based on technical correctness alone, while the novelty discussion should have been made with a consensus from the AC and the reviewers. I honestly tend to see this closer to a 4 than a 6, and I do not believe it would have increased.

`vCWP`: another critical reviewer of the novelty.

`vN4Q`: the only reviewer voting strongly in favor of acceptance and fully satisfied by the response. They would have been in a minority in a private discussion phase.

---

### Decision · Program_Chairs · 2026-01-26

Reject